# Long Noncoding RNA AROD Inhibits Host Antiviral Innate Immunity via the miR-324-5p–CUEDC2 Axis

Zixiao Zhang,[a] Tianqi Yu,[a] Haimin Li,[a] Liuyang Du,[a] Zian Jin,[a] Xiran Peng,[a] Yan Yan,[a] Jiyong Zhou,[a,b] Jinyan Gu[a]

[a]Key Laboratory of Animal Virology of Ministry of Agriculture, Zhejiang University Center for Veterinary Sciences, Hangzhou, China
[b]Collaborative Innovation Center and State Key Laboratory for Diagnosis and Treatment of Infectious Diseases, First Affiliated Hospital, Zhejiang University, Hangzhou, China

**ABSTRACT** Long noncoding RNAs (lncRNAs) are a class of noncoding RNAs that are involved in multiple biological processes. Here, we report a mechanism through which the lnc-AROD–miR-324-5p–CUEDC2 axis regulates the host innate immune response, using influenza A virus (IAV) as a model. We identified that host lnc-AROD without protein-coding capability is composed of 975 nucleotides. Moreover, lnc-AROD inhibited interferon-$\beta$ expression, as well as interferon-stimulated genes ISG15 and MxA. Furthermore, *in vivo* assays confirmed that lnc-AROD overexpression increased flu virus pathogenicity and mortality in mice. Mechanistically, lnc-AROD interacted with miR-324-5p, leading to decreased binding of miR-324-5p to CUEDC2. Collectively, our findings demonstrated that lnc-AROD is a critical regulator of the host antiviral response via the miR-324-5p–CUEDC2 axis, and lnc-AROD functions as competing endogenous RNA. Our results also provided evidence that lnc-AROD serves as an inhibitor of the antiviral immune response and may represent a potential drug target.

**IMPORTANCE** lnc-AROD is a potential diagnostic and discriminative biomarker for different cancers. However, so far the mechanisms of lnc-AROD regulating virus replication are not well understood. In this study, we identified that lnc-AROD is downregulated during RNA virus infection. We demonstrated that lnc-AROD enhanced CUEDC2 expression, which in turn inhibited innate immunity and favored IAV replication. Our studies indicated that lnc-AROD functions as a competing endogenous RNA that binds miR-324-5p and reduces its inhibitory effect on CUEDC2. Taken together, our findings reveal that lnc-AROD plays an important role during the host antiviral immune response.

**KEYWORDS** lnc-AROD, innate immunity, miR-324-5p, CUEDC2

Because most studies have focused on the function of proteins, less is known about the role of RNA, including noncoding RNA (ncRNA) (1). High-throughput genome and transcriptome data have revealed that 98% of the human genome is transcribed into ncRNA, some of which is classified as long noncoding RNA (lncRNA) (2). lncRNA is over 200 nucleotides (nt) in length, widely transcribed in mammalian cells, and has no protein-coding potential (3, 4). To date, several viral infections have been reported to substantially alter lncRNA expression (i.e., hepatitis B virus [5], human foamy virus [6], rabies virus [7], porcine reproductive respiratory syndrome virus [8], H3N2 swine influenza virus [9], H5N1 avian influenza A virus [10], avian leukosis virus type J [11], Seneca valley virus [12], duck plague virus [13], and bovine viral diarrhea virus [14]). Increasing evidence has shown that lncRNAs can regulate gene expression in the nucleus (15, 16) as well as affect microRNA (miRNA) signal transduction or absorption in the cytoplasm (17–19). Additionally, some lncRNAs (e.g., UBAP1-AST6 and HOXB-AS3) encode short peptides and play a role in promoting or inhibiting lung cancer proliferation (20, 21).

The lncRNA-activating regulator of Dickkopf WNT signaling pathway inhibitor 1

Address correspondence to Jiyong Zhou, jyzhou@zju.edu.cn, or Jinyan Gu, gujinyan@zju.edu.cn.

The authors declare no conflict of interest.

(DDK1) lncRNA (lnc-AROD), a spliced and polyadenylated lncRNA, has been reported to positively regulate DKK1 transcription at the level of transcription elongation, as well as recruit the general transcription factor EBP1 to the DKK1 promoter (22). Some studies have further revealed that lnc-AROD is upregulated in different tumor tissues and cells and can promote head and neck squamous cell carcinoma by forming a ternary complex with HSPA1A and YBX1. In addition, lnc-AROD can enhance hepatocellular carcinoma proliferation by interacting with serine- and arginine-rich splicing factor 3 (SRSF3) to trigger pyruvate kinase M (PKM) switching from PKM1 to PKM2 (23–25). Furthermore, lnc-AROD may serve as a potential diagnostic and discriminative biomarker for colorectal cancer and hepatocellular carcinoma.

Recently, lncRNAs have been found to be involved in modulating viral infection and the innate immune response (26). lncRNA-Gas5 inhibits hepatitis C virus (HCV) infection through binding the HCV NS3 protein (27), and interferon (IFN)-inducible lnc-Lsm3b suppresses RIG-I activity during the late stage of the innate immune response (28). lncRNA NEAT1 enhances interleukin-8 transcription via directing SFPQ and regulating HIV-1 posttranscriptional expression (29). In addition, lncRNA NRAV affects influenza A virus (IAV) replication through the inhibition of IFN-stimulated gene (ISG) transcription (30). lnc-ISG20 has also been found to enhance ISG20 translation and affect IAV replication (31); however, our understanding of lncRNAs is limited, with the functions and characteristics of most lncRNAs remaining unknown.

In the present study, genome-wide profiling of lncRNA expression identified an IAV-dysregulated novel lncRNA, termed lnc-AROD, which was capable of favoring IAV replication. lnc-AROD was found to function as a negative regulator in host antiviral immunity by suppressing ISG15 and MxA production. Further data demonstrated that lnc-AROD acts as a competing endogenous RNA (ceRNA) to enhance the level of CUEDC2 expression through sponging with miR-324-5p. Our results revealed that lnc-AROD is a negative regulator of IFN-$\beta$ and ISG expression, establishing a critical role for this lnc-RNA in the host innate defense during IAV infection.

## RESULTS

**lncRNA expression profile in H1N1-infected cells.** High-throughput RNA sequencing (RNA-seq) was performed to determine the lncRNA expression profiles in H1N1 virus-infected A549 cells at 18 h postinfection (hpi). Based on the specific location in the genome, the inclusion of multiple exons, and a length greater than 200 nucleotides (nt), noncoding transcripts were filtered to identify the novel lncRNAs. Moreover, noncoding transcripts were classified by CPAT, PFAM, and CPC. As a result, 258 distinct lncRNA candidates were selected (Fig. 1A and B). Furthermore, we analyzed the composition of lncRNAs in terms of the corresponding genomic locations of transcripts. Newly identified lncRNAs were categorized into five groups: sense lncRNAs (78%), intronic lncRNAs (15%), intergenic lncRNAs (6%), antisense lncRNAs (0.3%), and bidirectional lncRNAs (0.3%) (Fig. 1C). In addition, these lncRNAs were widely distributed in all chromosomes. Chromosomes 1 and 2 exhibited a significant enrichment of lncRNAs, whereas few altered lncRNAs were located on chromosome Y (Fig. 1D).

**Identification and characterization of lnc-AROD without protein-coding ability.** To explore crucial lncRNAs involved in a viral infection, hierarchical clustering was performed to determine the lncRNA expression profiles in A549 cells infected with or without the H1N1 virus for 18 h. The level of lncRNA expression was significantly altered following H1N1 infection (Fig. 2A). The differentially expressed lncRNAs, including 735 upregulated and 174 downregulated lncRNAs (fold change of >2), were detected following viral infection and clustered by volcano plot (Fig. 2B). There were 13 candidate lncRNAs selected to validate A549 cells with and without H1N1 infection. Figure 2C shows that the reverse transcription-quantitative PCR (RT-qPCR) data for the 13 lncRNAs were in agreement with the RNA-seq results. Of the 13 identified lncRNAs, it has been reported that NRAV is significantly downregulated after influenza virus infection (30). Another study reported that influenza virus can induce the production of NEAT1 (29). Here, lncRNA-AROD attracted our attention. lnc-AROD (also named lnc-

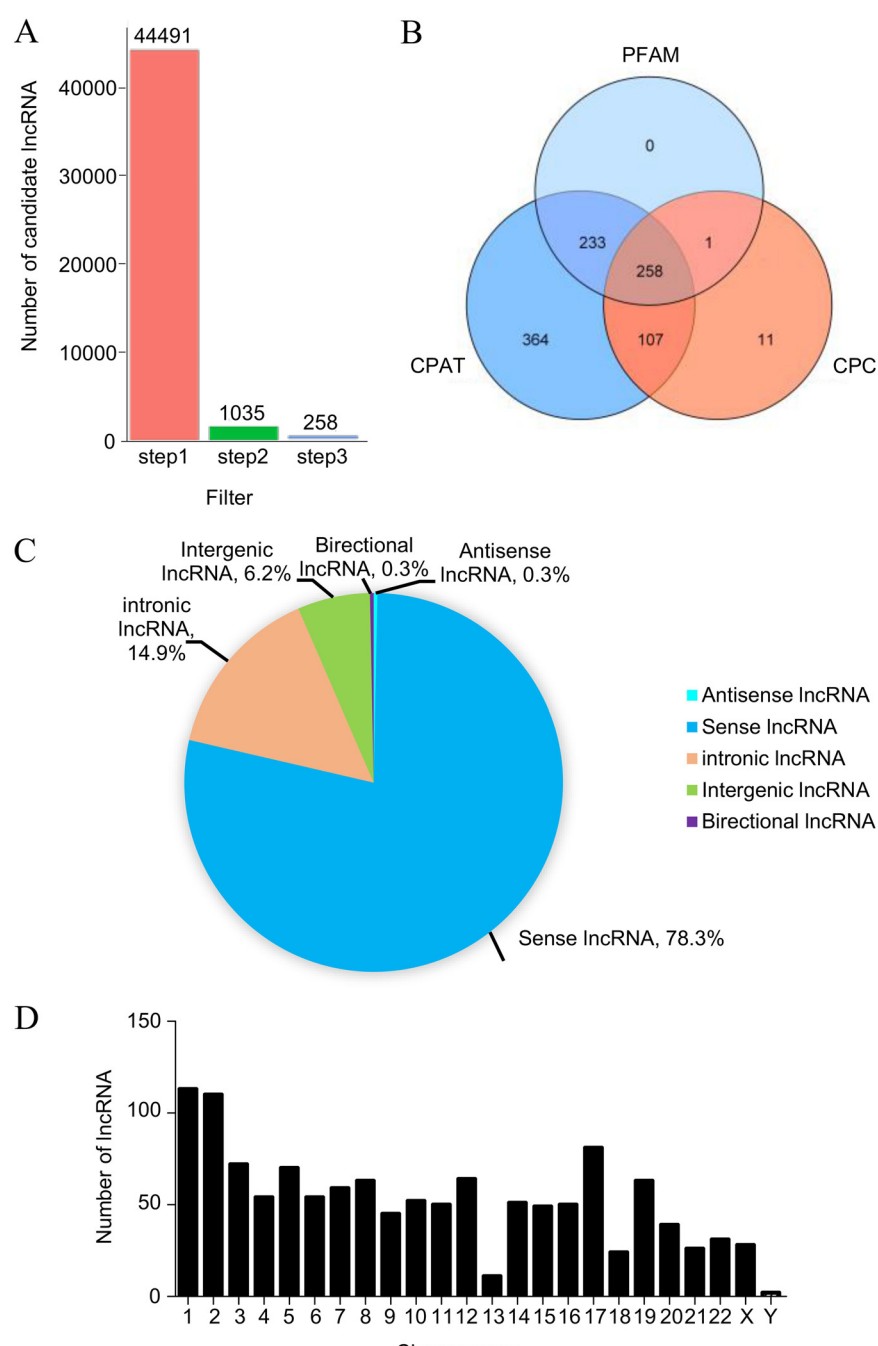

**FIG 1** Identification of novel lncRNAs in A549 cells following H1N1 infection. (A) lncRNA screening in H1N1-infected cells by filtration. (B) Evaluating the coding potential of assembled transcripts using three tools. (C) Classification of lncRNAs based on genomic location. (D) Distribution of differentially expressed lncRNAs in chromosomes.

MBL2-4 [25]), which consists of three exons and is located on chromosome 10q21.1, was significantly downregulated in the cells infected with different types of influenza A virus (Fig. 2D and E). We also found that lnc-AROD was in a time-dependent manner downregulated by IAV infection (Fig. 2F). Notably, vesicular stomatitis virus (VSV) and Sendai E virus (SeV) infections, but not herpes simplex virus (HSV) infection, could also downregulate lnc-AROD expression (Fig. 2G), indicating that RNA virus infection inhibited lnc-AROD expression. Subsequently, we determined the full lnc-AROD sequence using the 5′ and 3′ random amplification of cDNA ends (RACE) techniques. Figure 2H

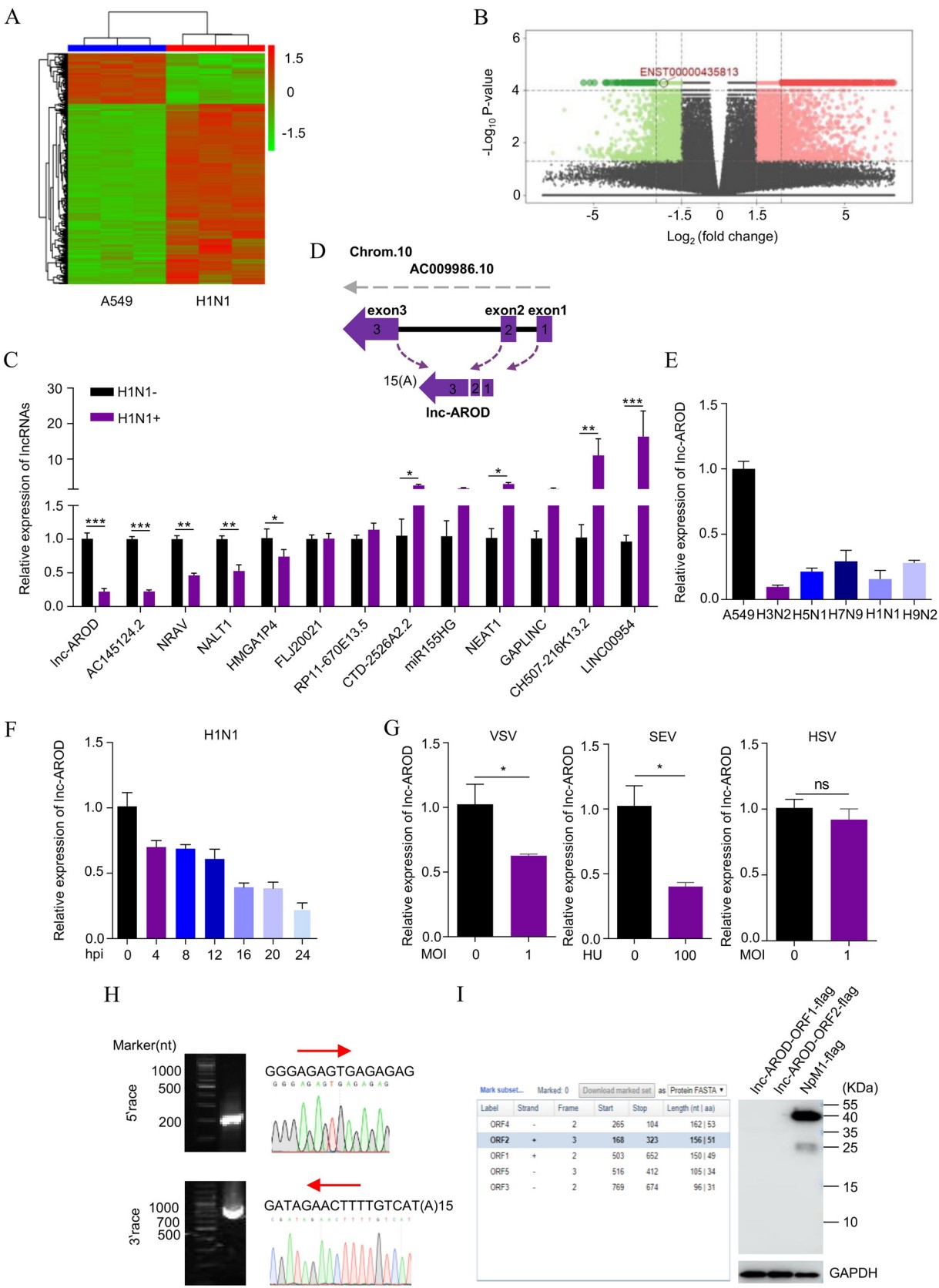

FIG 2 Characteristics of lnc-AROD in viral infection. (A) A clustered heatmap of differentially expressed lncRNAs was analyzed in H1N1-infected and uninfected A549 cells. (B) A volcano plot displaying 735 upregulated and 174 downregulated lncRNAs in H1N1-infected A549

shows that the full length of the lnc-AROD transcript was exactly 975 nt and contained a polyadenylated (15 As) tail. To verify whether lnc-AROD was indeed a noncoding RNA, we predicted two open reading frames (ORFs) of lnc-AROD by using ORF Finder (NCBI). We then constructed C-terminally FLAG-tagged lnc-AROD-ORF1–lnc-AROD-ORF2 and transfected them into 293T cells. Immunoblotting confirmed that the predicted ORF from lnc-AROD did not show its protein-coding ability (Fig. 2I). These data demonstrated that lnc-AROD is not a protein-coding lncRNA and that its expression is inhibited during RNA virus infection.

**lnc-AROD promotes IAV replication.** To determine the potential role of lnc-AROD during viral infection, we generated a lnc-AROD-overexpressing A549 cell line, termed lnc-AROD-A549 (Fig. 3A), and infected the cells with A/Puerto Rico/8/34 (H1N1) or A/Quail/Hangzhou/1/2013 (H9N2) as a virus model. RT-qPCR and immunoblotting revealed that the level of viral M mRNA and the NP and M1 viral proteins of the H1N1 and H9N2 viruses were higher in lnc-AROD-A549 cells than in uninfected control cells (Fig. 3B and C; see also Fig. S1A in the supplemental material). Furthermore, 50% tissue culture infective dose ($TCID_{50}$) assays revealed that the virus titers from lnc-AROD-A549 cells were significantly higher than those in the control cells (Fig. 3D and Fig. S1B). Next, we knocked down lnc-AROD in A549 cells by using RNA interference (si-lnc-AROD#1 and #2) (Fig. 3E) and infected the cells with the H1N1 and H9N2 viruses as described above. The data showed that the level of viral M mRNA and NP and M1 viral proteins of the H1N1 and H9N2 viruses in lnc-AROD knockdown cells were lower than those in the control cells (Fig. 3F and G and Fig. S1C). The viral titers from lnc-AROD knockdown cells were significantly lower than those in the control cells (Fig. 3H and Fig. S1D). Taken together, these findings suggested that lnc-AROD plays a positive regulatory role during IAV replication *in vitro*.

**lnc-AROD increases the pathogenicity of IAV in mice.** We established a mouse model system to verify the efficacy of lnc-AROD on IAV infection *in vivo*. Considering that lnc-AROD is not detected in mice, recombinant adeno-associated virus 6 (AAV6) expressing lnc-AROD or green fluorescent protein (GFP) were generated to express human lnc-AROD in mice. AAV–lnc-AROD and AAV-GFP (control) were intranasally administered to mice for 21 days, followed by the delivery of phosphate-buffered saline (PBS) or H1N1 virus ($10^{3.5}$ $TCID_{50}$) on day 0 (Fig. 4A). The AAV–lnc-AROD–infected mice exhibited high lnc-AROD expression in lungs (Fig. 4B). As expected, during IAV infection, the AAV–lnc-AROD–infected mice lost more weight than the AAV-GFP-infected mice (Fig. 4C), and all AAV–lnc-AROD–infected mice with IAV infection died within 9 days postinfection (dpi); however, approximately 53.8% AAV-GFP-infected mice survived (Fig. 4D). Moreover, RT-qPCR and $TCID_{50}$ revealed that the IAV virus titer and level of M mRNA expression were higher in the lungs of AAV–lnc-AROD mice on 3 dpi compared to that in the lungs of AAV-GFP mice (Fig. 4E and F). Consistently, there were obvious gross lesions in the lungs of AAV–lnc-AROD–infected mice compared with AAV-GFP-infected mice (Fig. 4G). Pathologic examination with hematoxylin and eosin (H&E) staining displayed more severe lymphocytic infiltration, a reduction of the alveolar airspace, thickening of the alveolar wall, and an alveolar cavity filled with inflammatory cells in the lungs of AAV–lnc-AROD mice compared to that of the controls at day 3 post-IAV infection (Fig. 4H). Together, these results suggested that lnc-AROD expression increases pathogenicity in mice by promoting IAV replication.

**FIG 2** Legend (Continued)

cells compared to the uninfected cells. Cells infected with H1N1 were collected at 18 hpi. (C) The differential expression of 13 selected lncRNAs was verified by RT-PCR in H1N1-infected and uninfected A549 cells. lnc-AROD is indicated by a red rectangle. (D) A schematic diagram of the genomic location of lnc-AROD. (E) A549 cells were infected with A/Swine/Guangdong/02/2005 (H3N2), A/pika/Qinghai/BI/2007 (H5N1), human-infective A/Hangzhou/1/2013 (H7N9), A/Puerto Rico/8/34 (H1N1), or A/Quail/Hangzhou/1/2013 (H9N2) at an MOI of 10 for 24 h. Total RNA was extracted to perform RT-PCR of lnc-AROD. (F) A549 cells were infected with H1N1 at an MOI for 0, 4, 8, 12, 16, 20, and 24 h. The expression of lnc-AROD was quantified by RT-qPCR. (G) A549 cells were infected with VSV, SeV, or HSV at the indicated doses for 24 h for lnc-AROD quantification via RT-qPCR. (H) The 5′ and 3′ end sequences of lnc-AROD in A549 cells were determined by RACE. Images of the RACE PCR products (left) and sequencing data (right) for lnc-AROD are shown. (I) Protein-coding potential was predicted in lnc-AROD by analysis using ORF Finder (left) and detected with a C-terminal Flag tag in all two reading frames by Western blotting (right). The results are expressed as means from three independent experiments. *, $P < 0.05$; **, $P < 0.01$; ***, $P < 0.001$.

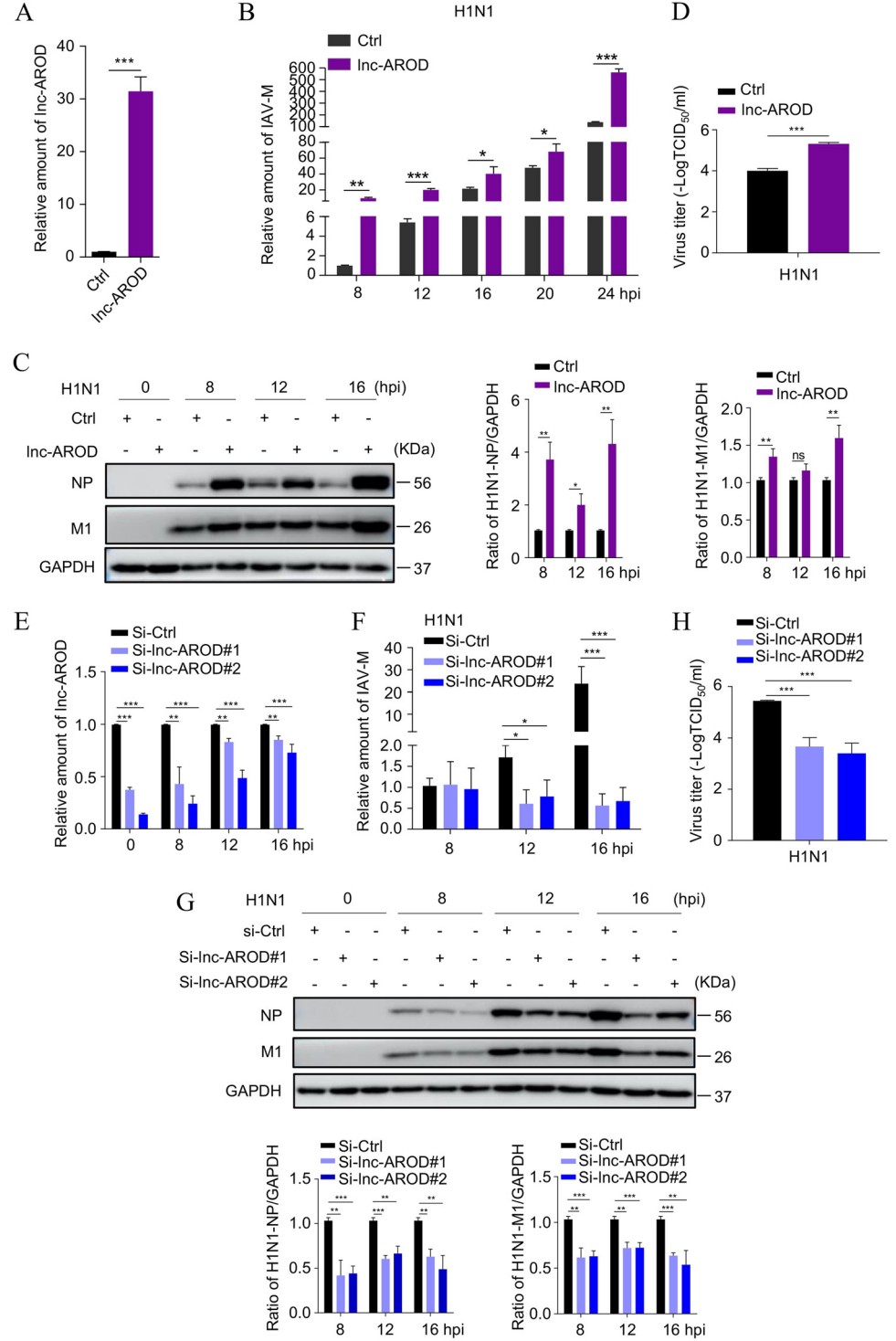

FIG 3 lnc-AROD promotes H1N1 virus replication. (A) The efficiency of lnc-AROD overexpression was determined by qRT-PCR in A549 cells. (B) The level of viral M mRNA expression was examined in lnc-AROD-overexpressing cells by qRT-PCR at the indicated time points. (C) The H1N1 virus NP and M1 proteins were separately measured in lnc-AROD-overexpressing cells by Western blotting. (D) The H1N1 virus titer was measured in lnc-AROD-overexpressing cells at 48 hpi. (E) The efficiency of lnc-AROD siRNA-based knockdown was quantified by qRT-PCR in infected and uninfected A549 cells. (F) The levels of viral M mRNA expression were examined in A549 cells transfected with lnc-AROD-specific siRNA by qRT-PCR at the indicated time points. (G) The H1N1 virus NP and M1 proteins were detected separately in A549 cells transfected with lnc-AROD-specific siRNA by Western blotting. (H) The H1N1 viral titer was measured in A549 cells transfected with lnc-AROD-specific siRNA at 48 hpi. The data are expressed as means from three independent experiments. *, $P < 0.05$; **, $P < 0.01$; ***, $P < 0.001$.

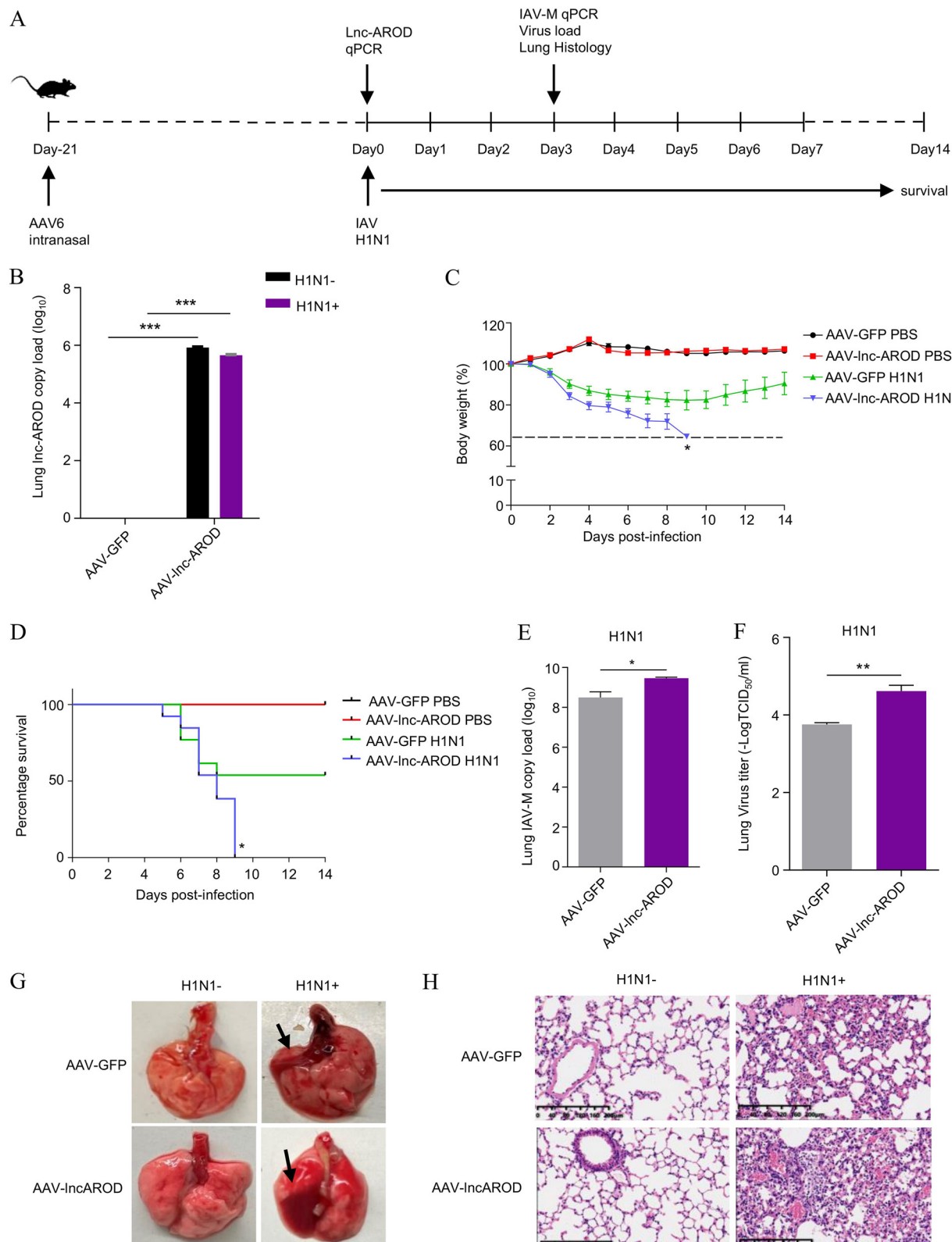

**FIG 4** lnc-AROD increases *in vivo* virulence of influenza A virus. (A) Strategy for the mouse experiment. Four-week-old C57BL/6 mice were intranasally administered AAV-GFP (control) or AAV–lnc-AROD (lnc-AROD) for 21 days. Eleven mice per group received PBS, and 16 mice per group were intranasally inoculated with H1N1 virus ($10^{3.5}$ $TCID_{50}$) on day 0. Three mice per group were euthanized on 3 dpi. Lungs were collected to determine the viral titer and histopathology. Body weight and survival were detected in the remaining mice until day 14 postinfection. (B and E) qRT-PCR for the abundance of lnc-AROD and IAV-M mRNA in mice lungs. (C and D) Infection kinetics in the mice

**lnc-AROD negatively regulates antiviral innate immunity.** To further understand the mechanism by which lnc-AROD enhances IAV replication, we employed high-throughput RNA-seq to profile the transcriptional response to H1N1 infection in lnc-AROD knockdown A549 cells. The data in Fig. 5A show the top 20 statistically significantly enriched KEGG pathways, suggesting that a lnc-AROD knockdown mainly interacted with an immune-associated pathway. Therefore, we next sought to determine whether lnc-AROD strengthens viral replication by interfering with the antiviral immune response. We first examined the effect of lnc-AROD on the expression of immune-related genes (the upstream genes of IFN-$\beta$, TBK1 and IRF3, and the downstream genes of IFN-$\beta$, ISG15 and MxA) in H1N1-infected A549 cells. The RT-qPCR data revealed that the knockdown of lnc-AROD significantly increased the level of IFN-$\beta$, ISG15, and MxA mRNA expression but not of TBK1 and IRF3 mRNA expression (Fig. 5B to G) following H1N1 infection. In contrast, lnc-AROD overexpression inhibited IFN-$\beta$, ISG15, and MxA expression levels in A549 cells after H1N1 infection (Fig. 5H to J). Similarly, lnc-AROD overexpression had no effect on the amount of TBK1 and IRF3 mRNA (Fig. 5K and L). We also measured the level of MxA protein expression in lnc-AROD-overexpressing cells and in A549 cells transfected with lnc-AROD-specific small interfering RNA (siRNA) by Western blotting. As shown in Fig. 5M, the overexpression of lnc-AROD reduced the amount of MxA protein in H1N1-infected A549 cells after H1N1 infection. Moreover, knockdown of lnc-AROD favored MxA protein expression upon H1N1 infection. Furthermore, ISG expression regulated by lnc-AROD was examined in IAV-infected AAV–lnc-AROD and AAV-GFP mice. Consistently, we found that the level of these ISGs in the AAV–lnc-AROD-infected mouse lungs were significantly reduced compared to those in AAV-GFP mice on 3 dpi (Fig. 5N). In summary, the data revealed that lnc-AROD functions as a negative modulator of antiviral innate immunity.

**lnc-AROD sponges with miR-324-5p.** lncRNAs have been shown to act as miRNA sponges to regulate various downstream targets (31, 32). Nuclear and cytoplasmic fractions, as well as an results of an *in situ* hybridization assay, indicated that the lnc-AROD concentration was not significantly decreased in the cytoplasm of cells with or without virus infection (Fig. 6A to C). To analyze the possible mechanism by which lnc-AROD inhibits antiviral innate immunity, we explored the ability of lnc-AROD to bind to miRNAs by using AGO2, an essential constituent of the RNA-induced silencing complex and an essential factor in the biological effect of miRNAs (33, 34). AGO2 RNA immunoprecipitation (RIP) was performed to check the abundance of lnc-AROD in lnc-AROD-293T cells expressing 3×Flag-AGO2 or 3×Flag-GFP. As shown in Fig. 6D, lnc-AROD was significantly enriched by 3×Flag-AGO2 compared to 3×Flag-GFP, indicating that lnc-AROD had potential miRNA sponge ability. Subsequently, the binding of four miRNAs to lnc-AROD, miR-149-5p, miR-214-5p, miR-324-5p, and miR1913 (Fig. 6E), was predicted by using RNAhybrid (35), starBase v2.0 (36), TargetScan (37), and miRanda (38). Next, we constructed a pGLO–lnc-AROD luciferase reporter plasmid and cotransfected HEK-293T cells with each of the predicted miRNA mimics. As shown in Fig. 6F, hsa-miR-324-5p was observed to inhibit pGLO-lnc-AROD luciferase reporter activity. Next, we generated a luciferase reporter plasmid containing the putative miR-324-5p binding site from lnc-AROD as well as a lnc-AROD mutant construct in which the miR-324-5p binding site was deleted. Cotransfection with miR-324-5p decreased the relative luciferase activity of lnc-AROD but had no impact on the lnc-AROD mutant construct (Fig. 6G and H). To further confirm the direct interaction between lnc-AROD and miR-324-5p, biotin-labeled positive and negative miR-324-5p probes were synthesized and used to test whether the probes could pull down lnc-AROD. The results of the qPCR analysis revealed that lnc-AROD could be pulled down by biotin-labeled miR-324-5p (Fig. 6I). Moreover, MS2-RIP assays confirmed that lnc-AROD could bind to miR-324-5p

**FIG 4** Legend (Continued)

were determined by body weight loss (C) and the survival curve (D). (F) The viral loads in the lungs of AAV-GFP and AAV–lnc-AROD mice were measured by TCID$_{50}$ assay on day 3 after H1N1 infection. (G) Parenchymal lung lesions were captured at 3 dpi. (H) Pathological slicing of the mouse lungs were stained with H&E. Scale bars, 200 $\mu$m. The data are expressed as means $\pm$ standard errors of the means. *, $P <$ 0.05; **, $P <$ 0.01.

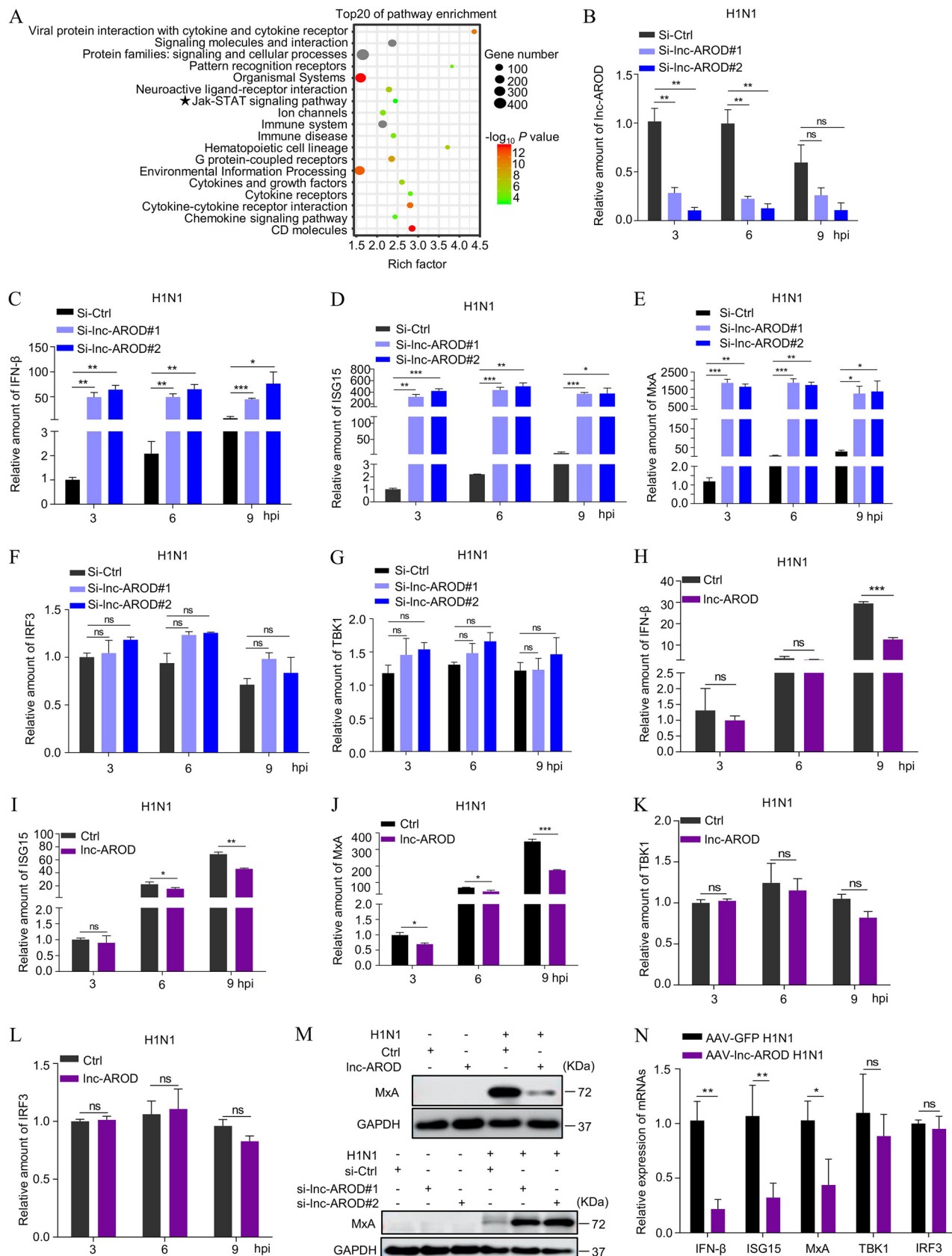

**FIG 5** lnc-AROD inhibits antiviral innate immunity. (A) KEGG analyses of the top 20 significantly enriched pathways of the target genes of an lnc-AROD siRNA-based knockdown following H1N1 infection. (B) The efficiency of an lnc-AROD siRNA-based knockdown was quantified by qRT-PCR in

(Fig. 6J). Moreover, a fluorescence *in situ* hybridization (FISH) assay also demonstrated that lnc-AROD interacted with miR-324-5p (Fig. 6K). Together, these results indicated that lnc-AROD may sponge with miR-324-5p.

**miR-324-5p regulates IAV replication via targeting the CUEDC2 3′-UTR.** It has been widely accepted that miRNAs can posttranscriptionally regulate target mRNA expression by binding to their 3′-untranslated region (UTR). Thus, we selected the predicted CUEDC2 as a potential target gene of miR-324-5p on TargetScan and miRanda. To determine whether the lnc-AROD–miR-324-5p axis involves CUEDC2 during IAV replication, we examined CUEDC2 expression after virus infection. The results from the qPCR assay showed that IAV infection could stimulate CUEDC2 downregulation (Fig. 7A). The sequence alignment showed that miR-324-5p had a complementary sequence with the 3′-UTR of CUEDC2 (Fig. 7B). To determine the interaction between miR-324-5p and CUEDC2–3′-UTR, the CUEDC2–3′-UTR and CUEDC2–3′-UTR binding site-deleted mutant (CUEDC2–3-UTR-del) were cloned into the luciferase reporter vector, pGLO, and cotransfected with miR-324-5p or control mimics. The results showed that the miR-324-5p mimics markedly inhibited luciferase activity when the wild-type 3′-UTR was transfected, whereas the deleted form had no response to miR-324-5p mimics (Fig. 7C). To confirm the direct interaction between miR-324-5p and CUEDC2, biotin-labeled miR-324-5p was used to test whether the probes could pull down CUEDC2. The results from the qPCR analysis revealed that CUEDC2 could be pulled down by biotin-labeled miR-324-5p in A549 and 293T cells (Fig. 7D). To test whether miR-324-5p participates in the regulation of CUEDC2 expression, we transfected miR-324-5p mimics or inhibitors into A549 cells. The Western blotting results confirmed that transfection with the miR-324-5p mimics but not miR-324-5p inhibitors suppressed CUEDC2 expression (Fig. 7E). Further detection revealed that the levels of IFN-$\beta$, ISG15, and MxA gene expression were significantly up-regulated in A549 cells transfected with the hsa-miR-324-5p mimic following H1N1 infection (Fig. 7F to H). Consistently, the H1N1 viral titer was significantly decreased in A549 cells transfected with the hsa-miR-324-5p mimic but not with the hsa-miR-324-5p inhibitor following H1N1 infection (Fig. 7I). The above results indicated that miR-324-5p inhibits CUEDC2 protein expression by targeting the CUEDC2–3′-UTR.

**lnc-AROD plays a ceRNA role in regulating CUEDC2 expression by binding to miR-324-5p.** The above data demonstrated that miRNA-324-5p directly interacts with lnc-AROD and CUEC2. Considering that lncRNA combines with miRNA as a ceRNA (31, 39), we wondered whether lnc-AROD functions as a ceRNA to regulate CUEDC2. The Western blotting results showed that the CUEDC2 expression was significantly downregulated in lnc-AROD knockdown cells (Fig. 8A); however, the qRT-PCR data revealed that CUEDC2 expression was upregulated in lnc-AROD-overexpressing cells, especially in lnc-AROD-overexpressing cells infected with H1N1, but not in lnc-AROD knockdown cells (Fig. 8B). These findings indicated that CUEDC2 expression is positively related to lncRNA expression. We subsequently further assessed the levels of IFN-$\beta$, ISG15, and MxA expression, as well as H1N1 virus replication, in CUEDC2-overexpressing and knockdown cells using RT-qPCR and TCID$_{50}$ assays. The data from Fig. 8C to H showed that the level of IFN-$\beta$, ISG15, and MxA expression was significantly downregulated in CUEDC2-overexpressing cells but not CUEDC2 knockdown cells infected with H1N1.

Consistently, the viral titer was significantly decreased in lnc-AROD-A549 cells with CUEDC2-specific siRNA transfection but not in CUEDC2-overexpressing cells after H1N1 infection, compared to that of control siRNA (Fig. 8I and J). Moreover, we also determined whether miR-324-5p could reverse the effect of lnc-AROD on the promotion of CUEDC2 expression. We cotransfected A549 cells with the pCDH-lnc-AROD plasmid and

**FIG 5** Legend (Continued)
H1N1 virus-infected cells. (C to G) The level of immune-related gene expression was examined in A549 cells transfected with lnc-AROD-specific siRNA by qRT-PCR at the indicated time points. (H to L) The level of immune-related gene expression was examined in lnc-AROD-overexpressing cells by qRT-PCR at the indicated time points. (M) The amount of MxA protein expression was separately measured in lnc-AROD-overexpressing and knockdown A549 cells by Western blotting. (N) The levels of *IFN-$\beta$*, *ISG15*, *MxA*, *IRF3*, and *TBK1* mRNA expression in AAV-lnc-AROD or AAV-GFP-infected mouse lungs were determined by qRT-PCR. The data are expressed as means of three independent experiments. *, $P < 0.05$; **, $P < 0.01$; ***, $P < 0.001$.

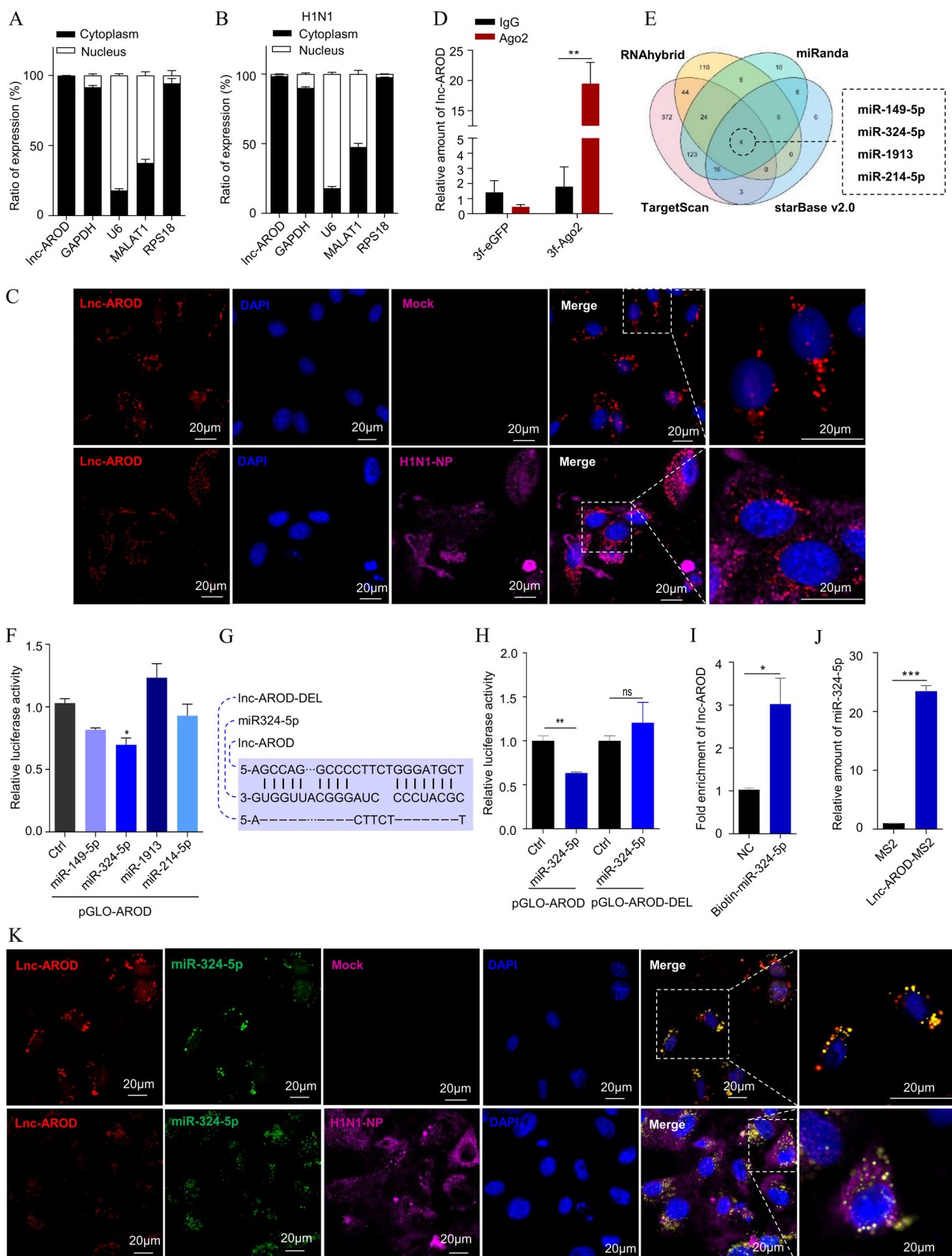

**FIG 6** lnc-AROD targets Hsa-miR-324-5p. (A) RT-qPCR analysis of lnc-AROD in the nuclear and cytoplasmic fractions of A549 cells. MALAT1, GAPDH, U6, and RPS18 mRNA expression levels were used as controls. (B) RT-qPCR analysis of lnc-AROD expression in the nucleus and cytoplasm of H1N1

different doses of an miR-324-5p mimic, followed by infection with the H1N1 virus. An immunoblotting assay showed that the expression levels of CUEDC2, NP, and M1 were depleted in a miR-324-5p dose-dependent manner in cells cotransfected with miR-324-5p and lnc-AROD, compared to that in cells transfected with lnc-AROD alone (Fig. 8K). These data suggested that miR-324-5p can hinder the upregulation of CUEDC2 expression favored by lnc-AROD (Fig. 9).

## DISCUSSION

lncRNA expression profiles have been found to display a large amount of lncRNA data. ln-AROD is upregulated in different tumor tissues and cells (23–25), and spliced and polyadenylated lncRNA positively regulates DKK1 transcription and recruits EBP1 to the DKK1 promoter (22). Conversely, our data showed that lnc-AROD is significantly downregulated in RNA virus-infected cells, demonstrating that lnc-AROD expression in RNA virus infection is different from that of tumors. Moreover, both *in vitro* and *in vivo* lnc-AROD overexpression, but not knockdown, facilitated RNA virus replication and not DNA virus replication. Notably, lnc-AROD overexpression significantly increased the mortality rate in mice during RNA virus infection. These findings suggested that lnc-AROD might be important for IAV pathogenesis *in vivo* and that it functions as a key factor via negatively regulating the expression of some crucial antiviral proteins.

miR-324-5p plays an important role in cancer, disease, and differentiation by interacting with noncoding RNA and proteins. miR-324-5p regulates mitochondrial morphology and cardiomyocyte cell death by targeting Mtfr1 (40). Moreover, CCAAT enhancer-binding protein delta was negatively regulated in papillary thyroid carcinoma via miR-324-5p targeting of protein tyrosine phosphatase receptor delta (41). Furthermore, miR-324-5p inhibited gallbladder carcinoma cell metastatic behaviors by downregulating transforming growth factor beta 2 expression (42). In addition, miR-324-5p directly targeted the 3′-untranslated region of NAD-dependent protein deacetylase sirtuin-1 (Sirt1) and negatively regulated the levels of Sirt1 in spinal cord injury (43). Moreover, miR-324-5p promoted mouse preadipocyte differentiation and increased fat accumulation in mouse fat targeting Krüppel-like factor 3 (44), inhibited C2C12 myoblast differentiation, and promoted intramuscular lipid deposition by targeting long noncoding Dum (lnc-Dum) and peptidase M20 domain containing 1 (Pm20d1) (45). Intriguingly, miR-324-5p has also been reported to target viral PB1 protein and host CUEDC2 to inhibit H5N1 replication (46). In our study, we found that among the target genes of miR-324-5p, CUEDC2 negatively regulated innate immunity, which was consistent with the function of lnc-AROD. Therefore, CUEDC2 was selected for subsequent research. Notably, our data revealed that miR-324-5p could directly bind to lnc-AROD and CUEDC2 and attenuated the upregulation of CUEDC2 expression enhanced by lnc-AROD in cells. These results demonstrated that lnc-AROD regulates CUEDC2 expression by directly targeting miR-324-5p. In general, our data suggested that lnc-AROD induces formation of the lnc-AROD–miR-324-5p–CUEDC2 axis by targeting miR-324-5p and functions as a ceRNA.

CUEDC2, a CUE-domain-containing protein that modulates inflammation, cell cycle, and tumorigenesis, can negatively regulate the JAK-STAT pathway to induce type I and type III interferons (46). In the present study, lnc-AROD, miR-324-5p, and CUEDC2 overexpression, but not knockdown, significantly decreased the level of IFN-$\beta$, ISG15, and MxA expression, respectively. Interestingly, lnc-AROD overexpression induced CUEDC2 upreg-

**FIG 6** Legend (Continued)

virus-infected A549 cells. (C) The cellular localization of lnc-AROD in A549 cells. The nuclei were stained with DAPI (blue), and the lnc-AROD probe was labeled with Cy3 (red). (D) AGO2 RIP was performed for detecting the amount of lnc-AROD in lnc-AROD-293T cells transfected with 3× Flag-AGO2 or 3× Flag-eGFP. IgG was used as the negative control. (E) RNAhybrid, starBase v2.0, TargetScan, and miRanda miRNA prediction programs consistently predicted four miRNAs interacted with lnc-AROD. (F) Luciferase activity of lnc-AROD in HEK293T cells transfected with miRNA mimics. (G and H) The plasmids pGLO-lnc-AROD and pGLO-lnc-AROD-del were generated to predict the miR-324-5p binding region. Luciferase activities of 293T cells transfected with hsa-miR-324-5p and pGLO-lnc-AROD or pGLO-lnc-AROD-del were quantified. (I and J) Bio-RNA pulldown and MS2-RIP assays were used to validate the interaction between hsa-miR-324-5p and lnc-AROD. (K) Colocalization between hsa-miR-324-5p and lnc-AROD was detected by RNA *in situ* hybridization in A549 cells. The data are expressed as means of three independent experiments. *, $P < 0.05$; **, $P < 0.01$; ***, $P < 0.001$.

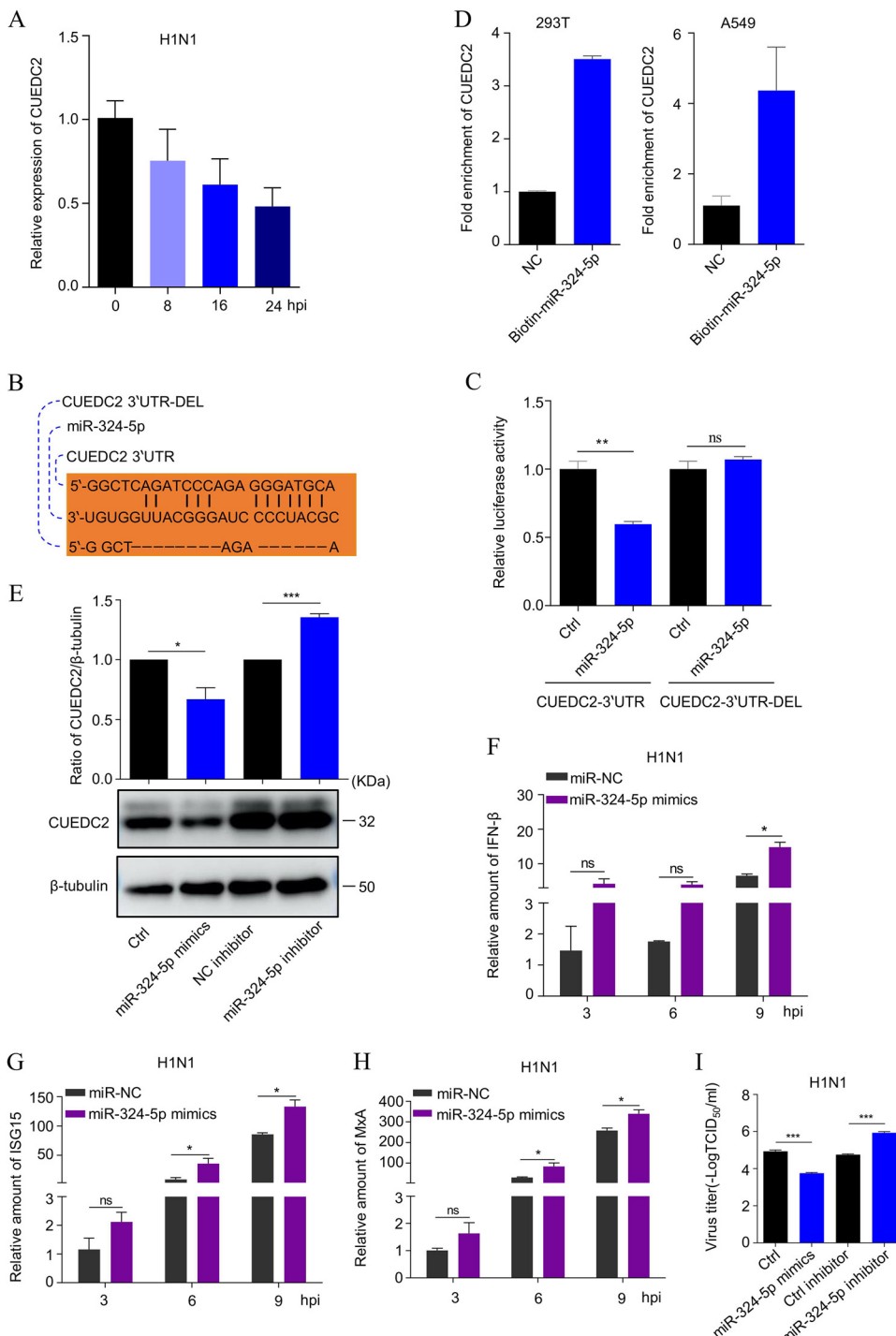

**FIG 7** The hsa-miR-324-5p targets CUEDC2–3′-UTR. (A) CUEDC2 expression in H1N1 virus-infected A549 or lnc-AROD cells. (B) Complementary sequence alignment of miR-324-5p with the 3′-UTR of CUEDC2 and CUEDC2–3′-UTR-Del. (C) The pGLO-CUEDC2–3′-UTR and pGLO-CUEDC2–3′-UTR-Del plasmids were generated for predicting the miR-324-5p-binding region. Luciferase activity of 293T cells transfected with hsa-miR-324-5p and pGLO-CUEDC2–3′-UTR or pGLO-CUEDC2–3′-UTR-Del were quantified. (D) A Bio-RNA pull down was used to detect the interaction between hsa-miR-324-5p and CUEDC2 in A549 and 293T cells, respectively. (E) The amount of CUEDC2 was measured in A549 cells transfected with control, hsa-miR-324-5p, control inhibitor, or hsa-miR-324-5p inhibitor by Western blotting. (F to H) The levels of IFN-$\beta$, ISG15, and MxA gene expression were examined in A549 cells transfected with the hsa-miR-324-5p mimic following H1N1 infection by qRT-PCR at the indicated time points. (I) The H1N1 virus titer was separately measured in A549 cells transfected with a hsa-miR-324-5p mimic and hsa-miR-324-5p inhibitor after H1N1 viral infection by TCID$_{50}$ assay. The data are expressed as means from three independent experiments. *, $P < 0.05$; **, $P < 0.01$; ***, $P < 0.001$.

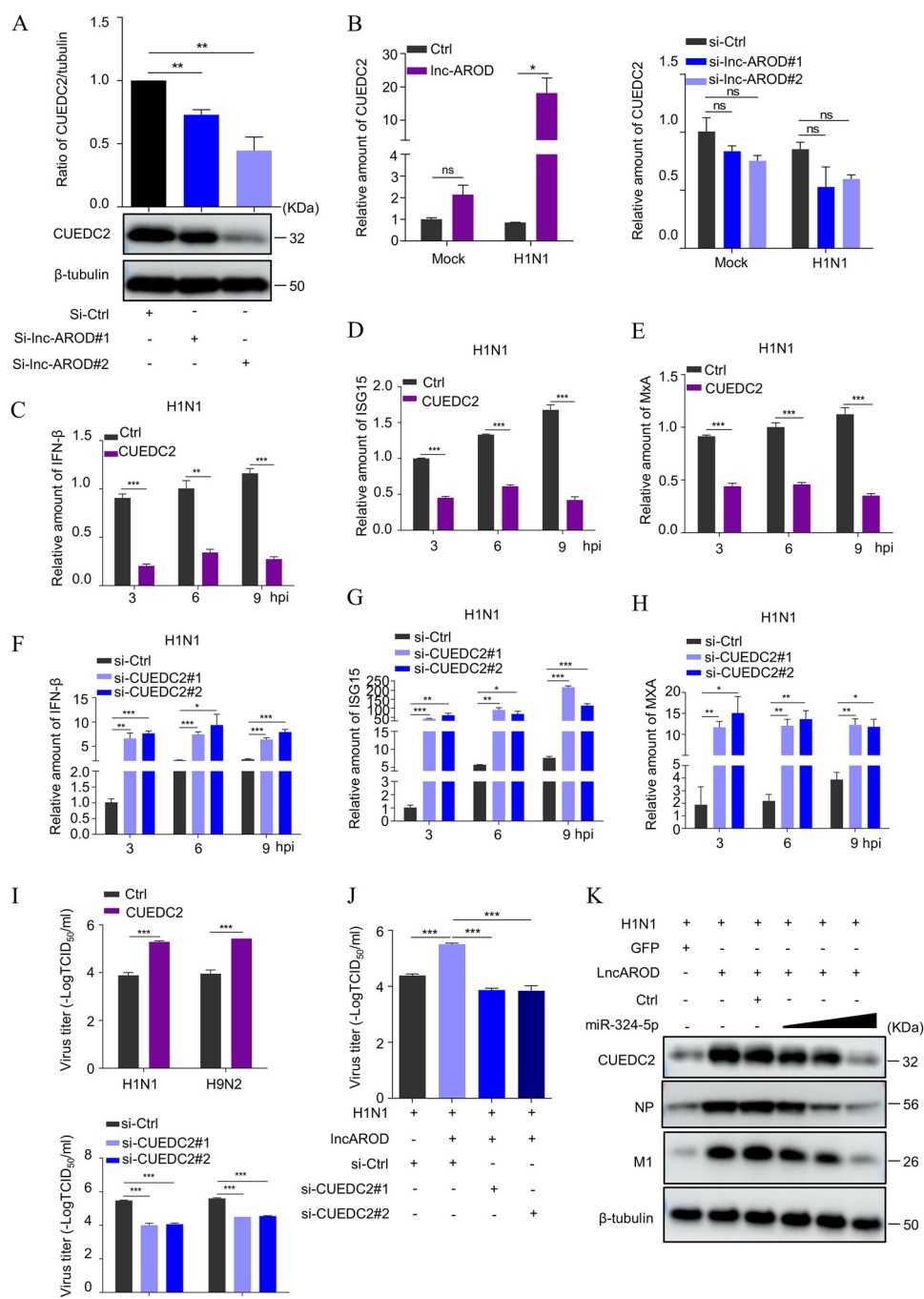

**FIG 8** lnc-AROD facilitates IAV replication via competitively binding to miR-324-5p with CUEDC2. (A) The level of CUEDC2 protein expression was measured in A549 cells transfected with lnc-AROD-specific siRNA by Western blotting. (B) The level of CUEDC2 mRNA expression was examined separately in lnc-AROD-overexpressing cells and in A549 cells transfected with lnc-AROD-specific siRNA by qRT-PCR. (C to H) The level of IFN-$\beta$, ISG15, and MxA gene expression was examined in A549 cells transfected with PCDH-CUEDC2 and CUEDC2-specific siRNA by qRT-PCR after virus infection at the indicated time points. (I) The H1N1 viral titer was measured in A549 cells transfected with the pCDH-CUEDC2 plasmid and CUEDC2-specific siRNA (si-CUEDC2#1 and #2) by TCID$_{50}$ assay at 48 hpi. (J) The H1N1 virus titer was detected in lnc-AROD-overexpressing cells following transfection with CUEDC2-specific siRNA or control siRNA by TCID$_{50}$ and Western blotting. (K) The level of H1N1 virus NP and M1 proteins and CUEDC2 protein expression were detected in A549 cells after cotransfection with lnc-AROD-overexpressing plasmid, control mimics, or different concentrations of miR-324-5p mimics (50, 100, or 200 pmol) by Western blotting. The data are expressed as means from three independent experiments. *, $P < 0.05$; **, $P < 0.01$; ***, $P < 0.001$.

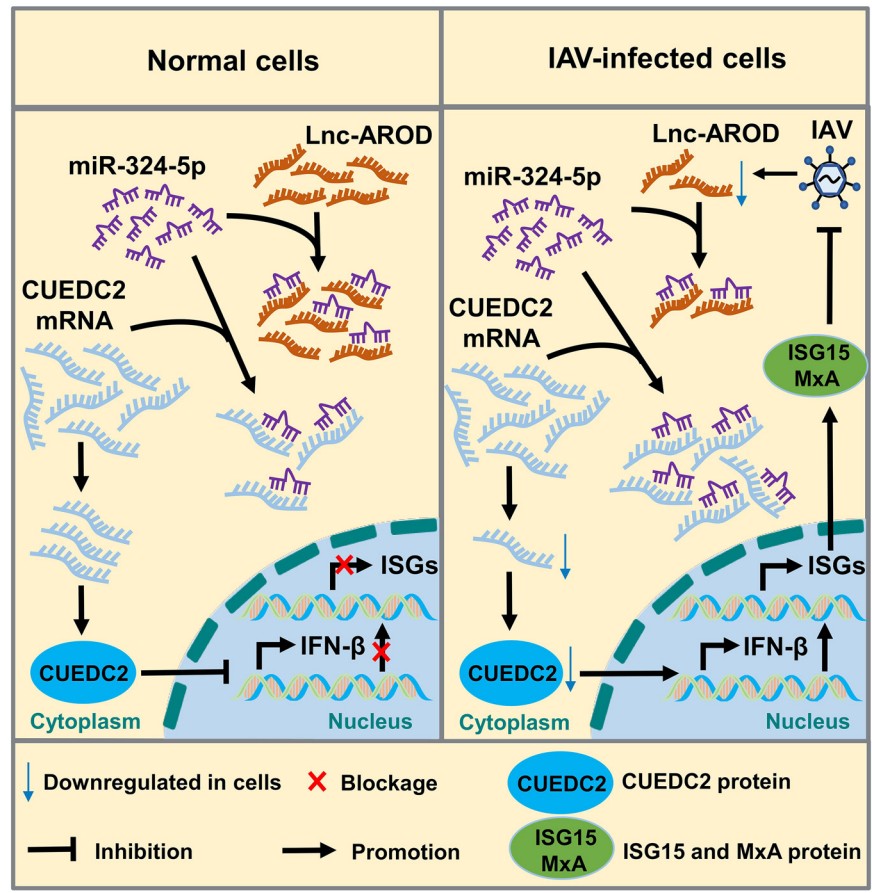

**FIG 9** Schematic representation of the proposed mechanism of lnc-AROD in IAV replication. Down-regulated lnc-AROD decreases miR-324-5p adsorption. Subsequently, free miR-324-5p binds directly to CUEDC2 and results in decreased free CUEDC2 mRNA, due to an increased miR-324-5p–CUEDC2 interaction complex, which triggers IFN-$\beta$, ISG15, and MxA transcription, which favors IAV replication.

ulation, and miR-324-5p overexpression induced decreased CUEDC2. Consequently, the overexpression of lnc-AROD and CUEDC2 favored viral replication, whereas miR-324-5p overexpression inhibited viral replication, demonstrating that lnc-AROD and CUEDC2, but not miR-324-5p, contribute to IAV replication. However, IAV infection exhibited downregulation of lnc-AROD expression and increased miR-324-5p binding to CUEDC2, suggesting that lnc-AROD regulates the level of IFN and ISG expression in a miR-324-5p- and CUEDC2-dependent manner. Taken together, these findings indicate that lnc-AROD positively regulates CUEDC2 expression by miR-324-5p and the lnc-AROD–miR-324-5p–CUEDC2 axis regulates host antiviral innate immunity.

## MATERIALS AND METHODS

**Plasmid construction, siRNAs, and miRNAs.** lnc-AROD, CUEDC2, and lnc-AROD-ORF1/ORF2-FLAG were cloned into the pCDH-CMV-MCS-EF1-copGFP plasmid (P0268; MiaolingBio, Wuhan, China); these constructs were named pCDH-AROD, pCDH-ORF1, pCDH-ORF2, and pCDH-CUEDC2. The full-length lnc-AROD, lnc-AROD-del, CUEDC2–3′-UTR, and CUEDC2–3′-UTR-del were inserted into the pGLO plasmid (Promega, Madison, WI, USA) and were named pGLO-AROD, pGLO-AROD-DEL, pGLO-CUEDC2–3′-UTR, and pGLO-CUEDC2–3′-UTR-del. Argonaute2 (AGO2) was cloned into the PCDH-3flag-EGFP plasmid (catalog number 21538; Addgene), and lnc-AROD was cloned into the pMS2 plasmid, which was gifted from Myriam Gorospe (47). lnc-AROD-specific siRNAs and CUEDC2-specific siRNAs were designed by Thermo RNAi Designer and transfected into human lung adenocarcinoma epithelial (A549) cells using jetPRIME transfection reagent (Polyplus, France) according to the manufacturer's instructions. The siRNA sequences used in this study are listed in Table S1. The miRNA mimics or inhibitors were synthesized by GenePharma (Shanghai, China), and their sequences are listed in Table S1.

**Cells and viruses.** Human embryonic kidney HEK293T (293T) cells and A549 cells were obtained from the Shanghai Institute of Cell Biology, Chinese Academy of Sciences (Shanghai, China). lnc-AROD-

overexpressing 293T and A549 cell lines were produced by cotransfection with pCDH-AROD, the helper lentiviral packaging plasmids psPAX2.0 (catalog number 12260; Addgene) and pMD2.G (catalog number 12259; Addgene). All cells were cultured in Dulbecco's modified Eagle's medium (DMEM; Gibco, Carlsbad, CA) supplemented with 10% fetal bovine serum (FBS; Thermo Scientific, USA). Cells were incubated at 37°C in a humidified 5% $CO_2$ atmosphere.

Influenza virus A/Puerto Rico/8/34 (H1N1), A/Quail/Hangzhou/1/2013 (H9N2), A/Swine/Zhejiang/04/2009 (H3N2), A/Hangzhou/1/2013 (H7N9), A/pika/Qinghai/Bl/2007 (H5N1), and VSV were stored by our laboratory and propagated in 9-day-old embryonated eggs at 37°C. SeV and HSV were propagated in 293T cells.

**High-throughput RNA sequencing of lncRNAs.** A549 cells were infected with H1N1 at a multiplicity of infection (MOI) of 10 for 18 h. The total RNA was extracted by TRIzol reagent (Vazyme Biotech Company, Nanjing, China) according to the manufacturer's instructions. rRNAs were further removed using a ribo-zero-magnetic-kit (Epicentre, Madison, WI, USA). RNA-seq libraries were prepared with a TruSeq RNA LT sample prep kit v2 (Illumina, San Diego, CA, USA) and sequenced on the Illumina Hiseq 3000 platform (Illumina Inc., San Diego, CA) at Shanghai Genergy Biotech (Shanghai, China). Samples were divided into two groups: H1N1-infected and uninfected (mock) cells. Each group contained three biological duplicates.

**Bioinformatic analysis of lncRNAs.** Reads containing adaptors, low-quality, and poly(N) were filtered out by Trim Galore_v0.4.2 (http://www.bioinformatics.babraham.ac.uk/projects/). Fast QC_v0.11.5 software (http://www.bioinformatics.babraham.ac.uk/projects/fastqc/) was used for a quality control analysis. The remaining reads were mapped to a human reference genome (version *Homo_sapiens* GRCh38) with the STAR_v2.5.2b software suite (48). The transcripts were assembled with the mapped reads using Stringtie_v1.3.1 (49). The coding ability of the lncRNAs was predicted by the Pfam database (http://pfam.xfam.org/), coding potential calculator (50), and coding potential assessment tool (51). Finally, the differentially expressed lncRNAs in H1N1-infected and uninfected cells were analyzed using DESeq2 (52).

**RNA extraction and quantitative PCR.** Total RNA was isolated using TRIzol reagent (Vazyme, Nanjing, China). The cDNA was synthesized using a Thermo Scientific RevertAid first-strand cDNA synthesis kit (Thermo Fisher Scientific, USA) to quantify the amount of mRNA and lnc-AROD. Glyceraldehyde 3-phosphate dehydrogenase (GAPDH) was used as an internal control. The cDNA was synthesized using a miRNA first-strand cDNA synthesis kit (by stem-loop; Vazyme, Nanjing, China) to quantify the amount of miRNA. Small nuclear U6 RNA was used as an internal standard. Real-time PCR analyses were performed using ChamQ Universal SYBR qPCR master mix (Vazyme, China). To determine the absolute quantity of RNA, the purified PCR product amplified from the cDNA corresponding to the lnc-AROD and IAV-M sequences was serially diluted to generate a standard curve. The primers used in this study are listed in Table S2.

**RACE.** The 5' and 3' RACE procedures were performed using a SMARTer RACE cDNA amplification kit (Clontech, Palo Alto, CA) in accordance with the manufacturer's protocol. The primers for lnc-AROD are listed in Table S1. The RACE PCR products were inserted into the pMD 18-T simple vector (TaKaRa, Japan) and sequenced.

**Dual luciferase assay.** The 293T cells were seeded into 24-well plates and transfected with pGLO-AROD, pGLO-AROD-DEL, pGLO-CUEDC2–3'-UTR, or pGLO-CUEDC2–3'-UTR-DEL together with miR-324-5p mimics. After 48 h, the luciferase activity was measured with a dual luciferase reporter gene assay kit (Beyotime, China) according to the manufacturer's protocol. Relative luciferase activity (firefly luciferase activity divided by *Renilla* luciferase activity) was calculated.

**RNA immunoprecipitation.** For AGO2 immunoprecipitation, lnc-AROD-overexpressing 293T cells were plated into six-well plates in three replicates and transfected with 3×Flag-AGO2 or 3×Flag-GFP for 48 h. The cells were UV cross-linked and lysed with RIPA buffer (10 mM HEPES [pH 7.4], 200 mM NaCl, 30 mM EDTA, and 0.5% Triton X-100) supplemented with RNase (Beyotime, China) and protease inhibitors for 15 min. The lysates were incubated with magnetic beads conjugated with IgG (as a control) and an anti-Flag antibody (F1804; Sigma-Aldrich, St. Louis, MO, USA) at 4°C for 4 h. The beads were washed five times and lysed in TRIzol reagent for RNA extraction and subjected to RT-qPCR to determine the amount of lnc-AROD.

An MS2-RIP assay was performed as previously reported (47). Briefly, 293T cells were plated into six-well plates in three duplicates and transfected with MS2 or MS2–lnc-AROD together with MS2-GST and miR-324-5p for 48 h. The cells were subsequently lysed with NT2 buffer (50 mM Tris-HCl [pH 7.0], 150 mM NaCl, 1 mM $MgCl_2$, 0.05% NP-40, 1 mM phenylmethylsulfonyl fluoride, 10 mM ribonucleoside vanadyl complex) supplemented with RNase (Beyotime, China) and protease inhibitors for 15 min. The lysates were incubated with GST-magnetic beads at 4°C for 4 h and washed five times, and RNA was extracted from the beads followed by RT-qPCR to quantify the amount of miR-324-5p.

**Biotin-coupled miRNA capture.** lnc-AROD-overexpressing 293T cells were seeded into six-well plates in three replicates and transfected with the 3' end of biotinylated miR-324-5p mimic or control RNA (BioSune, China) for 24 h, as previously described (53). Briefly, the cells were lysed with lysis buffer supplemented with RNase and protease inhibitors for 20 min. The remaining lysates were incubated with streptavidin-coupled magnetic beads (Life Technologies, USA) for 4 h at 4°C and washed five times with ice-cold lysis buffer. RNA was extracted from the remaining beads to evaluate the amount of lnc-AROD or CUEDC2 mRNA.

**RNA FISH.** The specific probe for the lnc-AROD sequence was synthesized by GenePharma (Shanghai, China), and miR-324-5p was transcribed to RNA fragments with a TranscriptAid T7 high-yield transcription kit (Thermo Scientific) and labeled with Alexa Fluor 488 using a ULYSIS nucleic acid labeling kit (Invitrogen, USA).

A549 cells were fixed with cold 4% paraformaldehyde for 30 min, incubated with 100% ethanol at 4°C for 12 h, and permeabilized with 0.5% Triton X-100 for 5 min. Wash buffer (nuclease-free water, 2× SSC [1× SSC is 0.15 M NaCl plus 0.015 M sodium citrate], 10% formamide) was added and subsequently incubated with hybridization buffer (nuclease-free water, 2× SSC, 50% formamide, 100 mg/mL dextran sulfate sodium salt) containing probe at 37°C for overnight. Nuclei were stained with 4′,6-diamidino-2-phenylindole (DAPI) and washed with PBS. Finally, images were obtained using a Zeiss LSM 880 laser confocal microscope. The probes used for FISH are listed in Table S1.

**Immunoblotting.** Cells were washed twice with ice-cold PBS and lysed in lysis buffer (5% sodium dodecyl sulfate [SDS], 1% Triton X-100, 50 mM Tris, 150 mM NaCl; pH 7.5). Samples were boiled and separated by 12% SDS-PAGE and transferred onto nitrocellulose membranes (Amersham Biosciences, NJ), followed by incubation with specific antibodies.

**Mouse experiments.** Four-week-old male C57BL/6 mice were purchased from Shanghai SLAC Animal Company (Shanghai, China) and maintained in specific-pathogen-free barrier facilities. A total of 54 mice were randomly divided into 4 groups: GFP + PBS group ($n = 11$); GFP–lnc-AROD + PBS group ($n = 11$); GFP + H1N1 group ($n = 16$); and GFP–lnc-AROD + H1N1 group ($n = 16$). To determine the level of lnc-AROD expression in the lungs, AAV-expressed lnc-AROD was produced by Vigene Biosciences, Inc. Briefly, lnc-AROD or negative sequences were inserted into an AAV6 vector, and the resultant vector was cotransfected with the helper packaging plasmids pAAV-RC and pHelper into 293T cells. After iodixanol gradient purification, the mice were intranasally administered approximately 25 $\mu$L (2.34 × $10^{13}$ v g/mL) AAV6-expressing lnc-AROD or negative control for 3 weeks. The mice were treated with 50 $\mu$L PBS or intranasally challenged with the H1N1 virus ($10^{3.5}$ $TCID_{50}$) to establish a mouse model of IAV infection.

**$TCID_{50}$ assay.** A549 cells were plated in 96-well plates and infected with serial dilutions of culture supernatant from cells or mouse lungs containing 1 $\mu$g/mL tosylsulfonyl phenylalanyl chloromethyl ketone–trypsin. At 48 hpi, a $TCID_{50}$ assay was carried out with an indirect immunofluorescence assay and calculated using the Reed-Muench method.

**Hematoxylin-eosin staining.** Briefly, the mouse lungs were removed and fixed in 4% polyformaldehyde at 3 dpi. Lung samples were embedded in paraffin and cut into 3-$\mu$m-thick sections, followed by an incubation with hematoxylin and eosin solution. The sections were mounted with coverslips and scanned. H&E staining was performed to observe the morphological changes in the lung tissues.

**Statistical analysis.** Statistical comparisons were analyzed using GraphPad Prism (version 5.0) software. All data are reported as means ± standard deviations (SD). The differences between groups were analyzed with a Student's $t$ test. $P$ values of <0.05 were considered statistically significant.

**Ethics statement.** The animal experiment was approved by the Institutional Animal Care and Use Committee (IACUC) of Zhejiang University (SYXK 2012-0178). All animal experiments were conducted in accordance with the Regulations for the Administration of Affairs Concerning Experimental Animals approved by the State Council of the Peoples Republic of China.

**Data availability.** The RNA sequencing data were deposited in the Gene Expression Omnibus database (accession number GSE208546).

## SUPPLEMENTAL MATERIAL

Supplemental material is available online only.
**SUPPLEMENTAL FILE 1**, PDF file, 0.05 MB.
**SUPPLEMENTAL FILE 2**, XLSX file, 0.01 MB.
**SUPPLEMENTAL FILE 3**, XLSX file, 0.01 MB.

## ACKNOWLEDGMENTS

This study is supported by grants from National Natural Foundation of China (grant number 32192454) and China Agriculture Research System (grant number CARS-40-K13). We thank Yan Yan for technical help from the MOA Key Laboratory of Animal Virology, Zhejiang University Center for Veterinary Sciences Center.

Z.Z., T.Y., J.G., and J.Z. designed the experiments. Z.Z., T.Y., and H.L. performed the experiments. Z.Z. and Z.J. performed the animal experiments. L.D., X.P., and Y.Y. provided experimental materials. Z.Z., J.G., and J.Z. analyzed the data. Z.Z., J.G., and J.Z. wrote the manuscript. Written consent for publication was obtained from all the participants.

We declare no competing interests.

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
