## [Reviewer comments · Microbiology Spectrum]

Microbiology Spectrum

Lnc-AROD inhibits host antiviral innate immunity via miR-324-5p/CUEDC2 axis

Jinyan Gu, Zixiao Zhang, Tianqi Yu, Haimin Li, Liuyang Du, Zian Jin, Xiran Peng, Yan Yan, and Jiyong Zhou

Corresponding Author(s): Jinyan Gu, Zhejiang University

Review Timeline:

Submission Date:	October 17, 2022
Editorial Decision:	February 2, 2023
Revision Received:	March 4, 2023
Accepted:	March 12, 2023

Editor: Bar-On Yotam

Reviewer(s): The reviewers have opted to remain anonymous.

Transaction Report:

DOI: <https://doi.org/10.1128/spectrum.04206-22>

February 2, 2023

Prof. Jiyong Zhou
Zhejiang University
Key Laboratory of Animal Virology of Ministry of Agriculture
Zhejiang University Center for Veterinary Sciences
Hangzhou
China

Re: Spectrum04206-22 (Lnc-AROD inhibits host antiviral innate immunity via miR-324-5p/CUEDC2 axis)

Dear Prof. Jiyong Zhou:

Only minor revision is needed.

In this study, the authors demonstrated an association between the abundance of lnc-AROD and the influenza A virus infection in both human cell lines and in the mouse model. W

Questions and minor revision suggestions for the authors:

1. In the experiments shown in Figure 5, A549 cells were first infected and then treated with lnc-AROD siRNAs. Why not the KD first and then infect the cells?
2. The rationale of why CUEDC2 is selected to be the interest of this study is missing in the manuscript. The authors wrote in the Discussion section about some published functions of CUEDC2, and its association with miR-324-5p. However, it was also mentioned in the Discussion section that CUEDC2 is not the only target of miR-324-5p. I believe the manuscript could benefit from an explanation of the authors' selection of genes of interest.
3. The illustration of the relationship between IAV infection, lnc-AROD abundance, and the associated change of CUEDC2 localization + downstream effect is somewhat misleading in Fig.9 and can be improved. The current expression in the figure is not the best.
4. LINE 47= Is there a reference. If so, please add reference in this point.

LINE 275= Further example should be added, together with other references.

Thank you for submitting your manuscript to Microbiology Spectrum. As you will see your paper is very close to acceptance. Please modify the manuscript along the lines I have recommended. As these revisions are quite minor, I expect that you should be able to turn in the revised paper in less than 30 days, if not sooner. If your manuscript was reviewed, you will find the reviewers' comments below.

When submitting the revised version of your paper, please provide (1) point-by-point responses to the issues raised by the reviewers as file type "Response to Reviewers," not in your cover letter, and (2) a PDF file that indicates the changes from the original submission (by highlighting or underlining the changes) as file type "Marked Up Manuscript - For Review Only". Please use this link to submit your revised manuscript. Detailed instructions on submitting your revised paper are below.

Link Not Available

Sincerely,

Bar-On Yotam

Reviewer comments:

Reviewer #1 (Comments for the Author):

Reviewer' Comments to Author:

The authors describe a mechanism through which the lnc22 AROD/miR-324-5p/CUEDC2 axis regulates the host innate immune response using 23 influenza A virus as a model.

From this article the conclusion is that lnc-AROD is a critical regulator of the host antiviral response via the miR-324-5p/CUEDC2 axis and lnc-AROD functions as competing endogenous RNA.

I do have some questions/suggestions to clarify the methods and about the interpretation of the results:

LINE 47= Is there a reference. If so, please add reference in this point.

LINE 275= Further example should be added, together with other references.

Reviewer #2 (Comments for the Author):

In this study, the authors demonstrated an association between the abundance of lnc-AROD and the influenza A virus infection in both human cell lines and in the mouse model. Within the manuscript, the authors claimed the following findings:

1. lnc-AROD interacts with miR-324-5p, and the interaction is potentially direct;
2. The above interaction leads to a decreased binding of miR-324-5p to the 3' UTR of CUEDC2 mRNA, which then causes an increased abundance of CUEDC2 protein;
3. lnc-AROD is a critical regulator of the host antiviral response (lnc-AROD abundance is positively associated with the virus titer post-IAV infection);
4. The antiviral function of lnc-AROD is realized through its direct interaction with miR-324-5p;
5. Through its regulatory effect on CUEDC2 via miR-324-5p, lnc-AROD abundance is negatively associated with the expression level of IFN- β , ISG15, and MxA.

The authors showed experimental evidence to support the above findings from the perspectives of both *ex vivo* and *in vivo*. To test the suggested molecular mechanism of action in regulating the innate immune response to IAV by lnc-AROD, the authors conducted both overexpression and knockdown studies using multiple experimental methodologies. They provided solid results to support the conclusion of this manuscript, as well as the suggested working model described in Fig. 9.

In Fig. 3&4, the authors showed that lnc-AROD abundance is associated with the infection rate of IAV, using both KD and overexpression of lnc-AROD. The experimental evidence was collected from both cultured cell lines and mouse models. Fig.5 showed that lnc-AROD has a suppressive effect on IFN- β , ISG15, and MxA, but not on IRF3 or TBK1. The authors should explain more on the rationale of the selection of genes being examined here in their study.

Through Fig. 6&7, the authors showed evidence to support the hypothesis that lnc-AROD can form a direct interaction with miR-324-5p, and this interaction "competes" the affinity between miR and CUEDC2 mRNA. The authors then suggested that this competition may lead to a modified/regulated expression of downstream genes that play important roles in innate immunity.

Moving on to Fig.8, the authors showed evidence that CUEDC2 itself has an effect on the expression level of the above-mentioned three genes that can be impacted by the level of lnc-AROD.

Moreover, the authors also presented data through the figures to show that viral titer can be impacted by the experimental treatment (such as overexpression of CUEDC2, etc), which tightly links the findings on the molecular level to the level of the biology of the cell line/animal model.

Questions and minor revision suggestions for the authors:

1. In the experiments shown in Figure 5, A549 cells were first infected and then treated with lnc-AROD siRNAs. Why not the KD first and then infect the cells?
2. The rationale of why CUEDC2 is selected to be the interest of this study is missing in the manuscript. The authors wrote in the Discussion section about some published functions of CUEDC2, and its association with miR-324-5p. However, it was also mentioned in the Discussion section that CUEDC2 is not the only target of miR-324-5p. I believe the manuscript could benefit from an explanation of the authors' selection of genes of interest.
3. The illustration of the relationship between IAV infection, lnc-AROD abundance, and the associated change of CUEDC2 localization + downstream effect is somewhat misleading in Fig.9 and can be improved. The current expression in the figure is not the best.

Reviewer #3 (Comments for the Author):

In their manuscript, Zhang et al employ genome wide RNA-seq profiling in IAV infected or non-infected 549 cell lines and identify several ncRNA that their transcription is changed upon infection. Among their candidates they identify AROD lncRNA and further show that it enhances IAV infection. AROD is defined as important for IAV pathogenesis *in vivo* and negatively regulates

the expression of ISGs.

Their data shows that AROD is downregulated in IAV RNA virus-infected cells. Moreover, both in vitro and in vivo AROD over-expression enhances replication of RNA viruses, while no effects were seen on DNA virus replication. They further show that in IAV infected mice, expression of AROD lncRNA via intranasal transduction of AAV, led to slight increased IAV titers and death of infected mice relative to AAV viruses expressing a control GFP gene.

Mechanistically they employ detailed in vitro and in vivo functional assays to demonstrate that AROD lncRNA suppresses ISG genes expression, defining AROD as a negative regulator of host innate response. Further analysis shows that AROD-lncRNA acts as a competitor RNA that enhances CUEDC2 expression via sponging miR-324-5p. Both CUEDC2 and miR-324-5p were nicely shown to associate with AROD and further manipulation of AROD lncRNA expression affected their expression. Overall, they conclude that miR-324-5p directly binds AROD and CUEDC2 to suppress the upregulation of CUEDC2 expression enhanced by AROD. Thus AROD regulates CUEDC2 expression by directly targeting miR-324-5p. and AROD induces formation of a complex with miR-324-5p and CUEDC2, targeting miR-324-5p and functioning as a ceRNA to overall suppress ISG expression.

Major remarks

1. It would be beneficiary that the screen for functional ncRNA will be conducted in primary target cells and not in a cell line. At least the validation of expression and effect on viral titer should be repeated in primary cells.
2. The authors show that KD of AROD increases levels of ISGs upon infection (Fig. 5)- No control is available of non-infected cells that either over-express or deplete of AROD lncRNA. What is the effect on ISGs expression in control mock cells, where AROD is KD or AROD over expressed; and in infected but not express AROD.
3. Nuclear and cytoplasmic fractions, as well as an in situ hybridization assays detected that the AROD concentration is not significantly decreased in the cytoplasm of cells with or without virus infection. Can the author monitor the levels in nuclear fractions. How does this reside with the overall decrease in AROD levels upon infection?).
4. Can the authors determine the actual copy number of AROD in each of their fractions using quantitative qPCR.
5. In Fig. 1 - please indicate if the enrichment is relative to non-infected cells. One can think to combine Fig 1+2.
6. Fig. 2E -a kinetic of lncRNA levels post infection is appropriate here.
7. Fig.2G - any information on AROD isoforms and the one that is most abundant in humans.
8. Fig3B - Effects are only shown for 16hpi. Can longer time frame be addressed.
9. Fig 3C+D - how significance are the infection differences upon over expressing AROD. It seems very low. Levels of M1 are not changing.
10. Fig 3H. - the effects of AROD silencing on infection is very subtle (x1.5fold). Same for the WB at panel G - differences in NP and M1 upon AROD silencing are very low. Maybe higher effects will be detected at later time points.
11. Fig. 4E - as noted above - the effects of AAV-AROD transduction on IAV titers are extremely low .
12. Fig. 7 - Can the authors also present data on ISG basal levels in non-infected cells (upon miR -mimics or inhibition).

Preparing Revision Guidelines

Please return the manuscript within 60 days; if you cannot complete the modification within this time period, please contact me. If you do not wish to modify the manuscript and prefer to submit it to another journal, please notify me of your decision immediately so that the manuscript may be formally withdrawn from consideration by Microbiology Spectrum.

If your manuscript is accepted for publication, you will be contacted separately about payment when the proofs are issued;

please follow the instructions in that e-mail. Arrangements for payment must be made before your article is published. For a complete list of **Publication Fees**, including supplemental material costs, please visit our website.

Reviewer' Comments to Author:

The authors describe a mechanism through which the lnc22 AROD/miR-324-5p/CUEDC2 axis regulates the host innate immune response using 23 influenza A virus as a model. From this article the conclusion is that lnc-AROD is a critical regulator of the host antiviral response via the miR-324-5p/CUEDC2 axis and lnc-AROD functions as competing endogenous RNA.

I due have some questions/suggestions to clarify the methods and about the interpretation of the results:

LINE 47= Is there a reference. If so, please add reference in this point.

LINE 275= Further example should be added, together with other references.

Response to Reviewers' comments:

Reviewer #1's comments:

The authors describe a mechanism through which the lnc-AROD/miR-324-5p/CUEDC2 axis regulates the host innate immune response using influenza A virus as a model. From this article the conclusion is that lnc-AROD is a critical regulator of the host antiviral response via the miR-324-5p/CUEDC2 axis and lnc-AROD functions as competing endogenous RNA.

I due have some questions/suggestions to clarify the methods and about the interpretation of the results:

1. LINE 47= Is there a reference. If so, please add reference in this point.

Response: thanks for your comments. Following your suggestions, a reference has been added to Introduction section of Revised Manuscript and highlighted in red words (lines 49 and 521-528).

2. LINE 275= Further example should be added, together with other references.

Response: thanks for your comments. According to your comment, two examples and corresponding references in this point have been added to Results section of Revised Manuscript and highlighted in red words (lines 277-281 and 624-632).

Reviewer #2's comments:

In this study, the authors demonstrated an association between the abundance of lnc-AROD and the influenza A virus infection in both human cell lines and in the mouse model. Within the manuscript, the authors claimed the following findings:

1. Lnc-AROD interacts with miR-324-5p, and the interaction is potentially direct;
2. The above interaction leads to a decreased binding of miR-324-5p to the 3' UTR of CUEDC2 mRNA, which then causes an increased abundance of CUEDC2 protein;
3. Lnc-AROD is a critical regulator of the host antiviral response (Lnc-AROD abundance is positively associated with the virus titer post-IAV infection);
4. The antiviral function of lnc-AROD is realized through its direct interaction with

miR-324-5p;

5. Through its regulatory effect on CUEDC2 via miR-324-5p, lnc-AROD abundance is negatively associated with the expression level of IFN- β , ISG15, and MxA.

The authors showed experimental evidence to support the above findings from the perspectives of both *ex vivo* and *in vivo*. To test the suggested molecular mechanism of action in regulating the innate immune response to IAV by lnc-AROD, the authors conducted both overexpression and knockdown studies using multiple experimental methodologies. They provided solid results to support the conclusion of this manuscript, as well as the suggested working model described in Fig. 9.

In Fig. 3&4, the authors showed that lnc-AROD abundance is associated with the infection rate of IAV, using both KD and overexpression of lnc-AROD. The experimental evidence was collected from both cultured cell lines and mouse models.

Fig.5 showed that lnc-AROD has a suppressive effect on IFN- β , ISG15, and MxA, but not on IRF3 or TBK1. The authors should explain more on the rationale of the selection of genes being examined here in their study.

Response: thanks for your comments. As we known, IRF3 and TBK1 are involved in the regulation of IFN- β and belong to the upstream genes of IFN- β , while ISG15 and MxA are interferon-inducible genes and belong to the downstream genes of IFN- β , and ISG15 and MxA are effectors of innate immunity. In our experimental results, lnc-AROD was involved in the regulation of IFN- β downstream genes (please see Figure 5C-E and 5H-J), therefore, IFN- β , ISG15 and MxA were selected for subsequent detection indicators in our study. And the explanation has been added in Results section of Revised Manuscript and highlighted in red words (lines 343-344).

Through Fig. 6&7, the authors showed evidence to support the hypothesis that lnc-AROD can form a direct interaction with miR-324-5p, and this interaction "competes" the affinity between miR and CUEDC2 mRNA. The authors then suggested that this competition may lead to a modified/regulated expression of downstream genes that play important roles in innate immunity.

Moving on to Fig.8, the authors showed evidence that CUEDC2 itself has an effect on the

expression level of the above-mentioned three genes that can be impacted by the level of lnc-AROD.

Moreover, the authors also presented data through the figures to show that viral titer can be impacted by the experimental treatment (such as overexpression of CUEDC2, etc), which tightly links the findings on the molecular level to the level of the biology of the cell line/animal model.

Questions and minor revision suggestions for the authors:

1. In the experiments shown in Figure 5, A549 cells were first infected and then treated with lnc-AROD siRNAs. Why not the KD first and then infect the cells?

Response: thanks for your comments. We deeply apologize for our incorrect description that led to your misunderstanding. We indeed knocked down lnc-AROD and then infected cells with virus in our experiments. The description about this experiments have been corrected in Result section of Revised Manuscript and highlighted in red words (line 338).

2. The rationale of why CUEDC2 is selected to be the interest of this study is missing in the manuscript. The authors wrote in the Discussion section about some published functions of CUEDC2, and its association with miR-324-5p. However, it was also mentioned in the Discussion section that CUEDC2 is not the only target of miR-324-5p. I believe the manuscript could benefit from an explanation of the authors' selection of genes of interest.

Response: thanks for your comments. In our functional study, we found that lnc-AROD played a negative regulator in innate immunity, and CUEDC2, the target gene of miR324-5p, could also negatively regulates immunity, which is consistent with the function of lnc-AROD. Therefore, we selected the CUEDC2 gene for subsequent research. And the explanation has been added in Discussion section of Revised Manuscript and highlighted in red words (lines 476-478).

3. The illustration of the relationship between IAV infection, lnc-AROD abundance, and the associated change of CUEDC2 localization + downstream effect is somewhat misleading in Fig.9 and can be improved. The current expression in the figure is not the best.

Response: thanks for your comments. According to your comment, we have modified the schematic diagram in Revised Figure 9.

Reviewer #3's comments:

In their manuscript, Zhang et al employ genome wide RNA-seq profiling in IAV infected or non-infected 549 cell lines and identify several ncRNA that their transcription is changed upon infection. Among their candidates they identify AROD lncRNA and further show that it enhances IAV infection. AROD is defined as important for IAV pathogenesis in vivo and negatively regulates the expression of ISGs.

Their data shows that AROD is downregulated in IAV RNA virus-infected cells. Moreover, both in vitro and in vivo AROD over-expression enhances replication of RNA viruses, while no effects were seen on DNA virus replication. They further show that in IAV infected mice, expression of AROD lncRNA via intranasal transduction of AAV, led to slight increased IAV titers and death of infected mice relative to AAV viruses expressing a control GFP gene.

Mechanistically they employ detailed in vitro and in vivo functional assays to demonstrate that AROD lncRNA suppresses ISG genes expression, defining AROD as a negative regulator of host innate response. Further analysis shows that AROD-lncRNA acts as a competitor RNA that enhances CUEDC2 expression via sponging miR-324-5p. Both CUEDC2 and miR-324-5p were nicely shown to associate with AROD and further manipulation of AROD lncRNA expression affected their expression. Overall, they conclude that miR-324-5p directly binds AROD and CUEDC2 to suppress the upregulation of CUEDC2 expression enhanced by AROD. Thus AROD regulates CUEDC2 expression by directly targeting miR-324-5p. and AROD induces formation of a complex with miR-324-5p and CUEDC2, targeting miR-324-5p and functioning as a ceRNA to overall suppress ISG expression.

Major remarks

1. It would be beneficiary that the screen for functional ncRNA will be conducted in primary target cells and not in a cell line. At least the validation of expression and effect on viral titer should be repeated in primary cells.

Response: thanks for your comments. According to your comment, we isolated and cultured

the primary lung fibroblasts from the fetal mice (MPF), and electrotransfected lnc-AROD to achieve overexpression of lnc-AROD in primary lung fibroblasts (Please see the following figure A), and then infected H1N1 virus to detect the effect on IAV replication. In viral titer assay, we found that compared with the control group, the H1N1 viral titer was significantly increased in primary mouse lung fibroblasts transfected with pCDH-AROD plasmid by about 10 times (Please see the following figure B). And combined with in vivo experiments in mice (please see Figure 4), the function of lnc-AROD has been determined.

Figure. lnc-AROD promotes H1N1 virus replication in primary lung fibroblasts. (A) The efficiency of lnc-AROD overexpression was determined by qRT-PCR in primary lung fibroblasts. (B) The H1N1 viral titer was measured in primary lung fibroblasts transfected with pCDH-lnc-AROD plasmid at 48 hpi.

2. The authors show that KD of AROD increases levels of ISGs upon infection (Fig. 5)- No control is available of non-infected cells that either over-express or deplete of AROD LncRNA. What is the effect on ISGs expression in control mock cells, where AROD is KD or AROD over expressed; and in infected but not express AROD.

Response: thanks for your comments. Following your suggestion, we detected the expression of ISGs in lnc-AROD-A549 cells, and we also knocked down lnc-AROD in uninfected A549 cells to detect the expression of ISGs. Meanwhile, we detected the levels of ISGs in infected

cells but not overexpressed or knocked down of lnc-AROD. In qPCR assays, we found that compared to the control cells, the expressions of ISGs had no significant change in cells that overexpressed or knocked down of lnc-AROD without virus infection (Please see the following figure A and B). And the results showed that the levels of ISGs gradually increased after virus infection (Please see the following figure C).

A

B

C

Figure. Detection on the levels of ISGs. (A) The levels of ISGs were examined in A549 cells transfected with lnc-AROD-specific siRNA by qRT-PCR without virus infection. (B) The levels of ISGs were examined in lnc-AROD-overexpressing cells by qRT-PCR without virus infection. (C) The levels of ISGs were examined in A549 cells by qRT-PCR at the indicated time points after H1N1 virus infection.

3. Nuclear and cytoplasmic fractions, as well as an *in situ* hybridization assays detected that the AROD concentration is not significantly decreased in the cytoplasm of cells with or without virus infection. Can the author monitor the levels in nuclear fractions. How does this reside with the overall decrease in AROD levels upon infection?).

Response: thanks for your comments. We selected three groups (group 1 from Figure 6C, group 2 from Figure 6K and group 3 from other images) of infected and uninfected lnc-AROD fluorescence images (Please see the following figure), and analyzed the mean fluorescence intensity of lnc-AROD in a single cell with or without virus infection. The results showed that the mean fluorescence intensity of lnc-AROD decreased significantly after H1N1 infection (Please see the following figure). According to your comment, we performed the absolute quantitative analysis of lnc-AROD for nuclear-cytoplasm separated samples (The results are presented in the next question). And we found that there was no significant cytoplasmic or nuclear transfer of lnc-AROD after virus infection.

B

Figure. Analysis of the mean fluorescence intensity of lnc-AROD in a single cell with or without virus infection. (A) Three groups of infected and uninfected lnc-AROD fluorescence images were selected to analyze the mean fluorescence intensity of lnc-AROD. (B) The mean fluorescence intensity of lnc-AROD in a single cell using Image J software.

4. Can the authors determine the actual copy number of AROD in each of their fractions using quantitative qPCR.

Response: thanks for your comments. According to your comment, we detected the copy number of lnc-AROD in the cytoplasm and nucleus of uninfected or infected A549 cells. In qPCR assays, the standard curve of lnc-AROD is shown below (Please see the following figure A). And the results showed that the copy number of lnc-AROD in cytoplasm was decreased after H1N1 infection, and there was no significant change in the nuclear fraction (Please see the following figure B). U6 and RPS18 were used as internal control of nucleus and cytoplasm, respectively (Please see the following figure C and D).

A

B

Figure. The copy number of AROD in cytoplasm and nucleus with or without virus infection were detected using qPCR. (A) The lnc-AROD copy number in A549 cells was determined by the standard curve. (B) The lnc-AROD copy number in cytoplasm and nucleus were examined by qPCR. (C and D) U6 and RPS18 were used as internal control of nucleus and cytoplasm, respectively.

5. In Fig. 1 - please indicate if the enrichment is relative to non-infected cells. One can think to combine Fig 1+2.

Response: thanks for your comments. Figure 1 is not the result of enrichment. Figure 1A and B are new lncRNAs identified in infected cells, Figure 1C and D are all lncRNAs identified for classification and chromosome distribution statistics. While, the results in the Heat Maps and Volcano Plot of Figure 2A and B are relative to non-infected cells. Therefore, we prefer to put Figure 1 and Figure 2 separately.

6. Fig. 2E -a kinetic of lncRNA levels post infection is appropriate here.

Response: thanks for your comments. Following your suggestion, we detected the expression of lnc-AROD in A549 cells at different time points after virus infection. The data showed that lnc-AROD was in a time-dependent manner downregulated by IAV infection. The data was exhibited in figure 2F of revised Figure 2, and the corresponding detail description and figure legend have been added in Results and Figure legends section of Revised Manuscript and

highlighted in red words (lines 283-284, lines 719-721). The number of the following figure was modified in revised Figure 2 and Manuscript.

7. Fig.2G - any information on AROD isoforms and the one that is most abundant in humans.

Response: thanks for your comments. AROD isoforms were not found in our high-throughput sequencing results, and was also proved by RACE experiments.

8. Fig3B - Effects are only shown for 16hpi. Can longer time frame be addressed.

Response: thanks for your comments. According to your comment, we examined the effect of overexpression of lnc-AROD on the level of viral M mRNA at 8h, 12h, 16h, 20h and 24h after H1N1 infection. The data showed that compared to the control cells, the level of viral M mRNA of H1N1 virus was higher in lnc-AROD-A549 cells. The data was updated and showed in figure 3B of revised Figure 3.

9. Fig 3C+D - how significance are the infection differences upon over expressing AROD. It seems very low. Levels of M1 are not changing.

Response: thanks for your comments. According to our grayscale analysis, the expression of M1 and NP were significantly improved in Figure 3C, the data were statistically significant. In the titer result of Figure 3D, the ordinate is the result of logarithm. Therefore, compared with the control group, overexpression of lnc-AROD can increase the virus titer of H1N1 by about 20 times. We also modified the ordinate character of the virus titer in Figure 3D (Please see Revised Figure 3).

10. Fig 3H. - the effects of AROD silencing on infection is very subtle (x1.5fold). Same for the WB at panel G - differences in NP and M1 upon AROD silencing are very low. Maybe higher effects will be detected at later time points.

Response: thanks for your comments. According to our grayscale analysis, the expression of M1 and NP were significantly decreased in Figure 3G, the data were statistically significant. In the titer result of Figure 3H, the ordinate is the result of logarithm. Therefore, compared with the control group, knockdown of lnc-AROD can decrease the virus titer of H1N1 by

about 100 times. We also modified the ordinate character of the virus titer in Figure 3H (Please see Revised Figure 3).

11. Fig. 4E - as noted above - the effects of AAV-AROD transduction on IAV titers are extremely low.

Response: thanks for your comments. In the titer result of Figure 4F, the ordinate is the result of logarithm. Therefore, IAV virus titer was about 8 times higher in the lungs of AAV-lnc-AROD mice compared to that in the lungs of AAV-GFP mice. We also modified the ordinate character of the virus titer in Figure 4F (Please see Revised Figure 4).

12. Fig. 7 - Can the authors also present data on ISG basal levels in non-infected cells (upon miR -mimics or inhibition).

Response: thanks for your comments. Following your suggestion, we overexpressed miR-324-5p mimics in uninfected cells to detect the expressions of ISGs. In qPCR assays, we found that compared to the control cells, the expressions of ISGs had no significant change in cells that overexpressed miR-324-5p without virus infection (Please see the following figure A).

Figure. Detection on the levels of ISGs. (A) The levels of ISGs were examined in in A549 cells transfected with miR-324-5p mimics by qRT-PCR without virus infection.

March 12, 2023

Dr. Jinyan Gu
Zhejiang University
Hangzhou
China

Re: Spectrum04206-22R1 (Lnc-AROD inhibits host antiviral innate immunity via miR-324-5p/CUEDC2 axis)

Dear Dr. Jinyan Gu:

Your manuscript has been accepted, and I am forwarding it to the ASM Journals Department for publication. You will be notified when your proofs are ready to be viewed.

Sincerely,

Bar-On Yotam
Editor, Microbiology Spectrum
